# CONSTRAINT-FREE STRUCTURE LEARNING WITH SMOOTH ACYCLIC ORIENTATIONS

**Riccardo Massidda, Francesco Landolfi, Martina Cinquini, Davide Bacciu**
Department of Computer Science
Università di Pisa, Italy
`{riccardo.massidda,francesco.landolfi,martina.cinquini}@phd.unipi.it`
`davide.bacciu@unipi.it`

## ABSTRACT

The structure learning problem consists of fitting data generated by a Directed Acyclic Graph (DAG) to correctly reconstruct its arcs. In this context, differentiable approaches constrain or regularize an optimization problem with a continuous relaxation of the acyclicity property. The computational cost of evaluating graph acyclicity is cubic on the number of nodes and significantly affects scalability. In this paper, we introduce COSMO, a constraint-free continuous optimization scheme for acyclic structure learning. At the core of our method lies a novel differentiable approximation of an orientation matrix parameterized by a single priority vector. Differently from previous works, our parameterization fits a smooth orientation matrix and the resulting acyclic adjacency matrix without evaluating acyclicity at any step. Despite this absence, we prove that COSMO always converges to an acyclic solution. In addition to being asymptotically faster, our empirical analysis highlights how COSMO performance on graph reconstruction compares favorably with competing structure learning methods.

## 1 INTRODUCTION

Directed Acyclic Graphs (DAGs) are a fundamental tool in several fields to represent probabilistic or causal information about the world (Koller & Friedman, 2009; Pearl, 2009). A fundamental problem in this context concerns the retrieval of the underlying structure between a set of variables, i.e., the problem of identifying which arcs exist between nodes associated to the variables of interest (Spirtes et al., 2000). In recent years, applications of structure learning to causal discovery led to growing interest in tackling the problem using gradient-based methods that optimize a smooth representation of a DAG (Vowels et al., 2022). For instance, while not suitable for causal discovery *per se* (Reisach et al., 2021), acyclic structure learners are fundamental components of most state-of-the-art continuous causal discovery algorithms (Lachapelle et al., 2020; Brouillard et al., 2020; Lorch et al., 2022). A well-established technique, popularized by NOTEARS (Zheng et al., 2018), consists of computing the trace of the matrix-exponential of the adjacency matrix, which is differentiable and provably zero if and only if the corresponding graph is acyclic. However, despite their widespread adoption, NOTEARS-like acyclicity constraints impose a cubic number of operations in the number of nodes per optimization step and substantially prevent scalable and applicable continuous discovery algorithms.

In this context, we propose a novel formulation and optimization scheme for learning acyclic graphs that avoids evaluating the acyclicity of the solution in any optimization step. Notably, our proposal does not sacrifice theoretical guarantees of asymptotic convergence to acyclic solutions which apply to existing structure learning methods (Ng et al., 2022a). At the core of our scheme lies a novel definition of *smooth* orientation matrix, i.e., a differentiable approximation of an orientation matrix parameterized by a priority vector on the graph nodes. The priority vector represents a discrete orientation where each node has an outgoing arc to all nodes with higher priority. We define our *smooth* orientation matrix by applying a tempered sigmoid to the pair-wise priority differences, which equals the discrete orientation in the limit of the sigmoid temperature to zero. By annealing temperature during training, we prove that we are effectively decreasing an upper bound on the acyclicity of the solution. Further, we show that the parameterization represents the space of DAGs as a differentiable function of a directed graph and our smooth orientation matrix. Since our approach

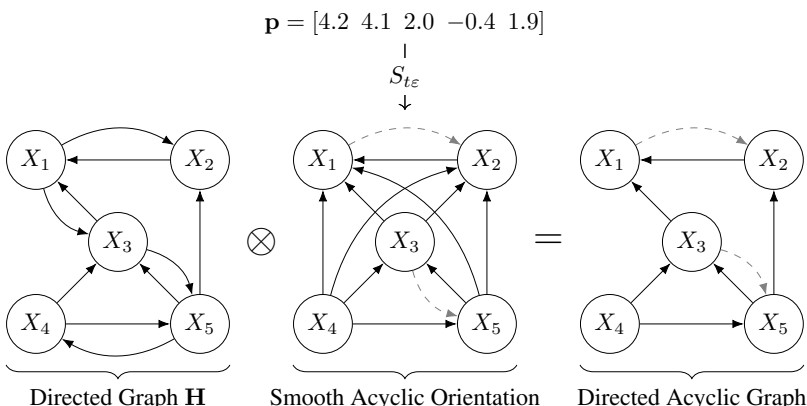

Figure 1: In COSMO we propose to optimize an acyclic adjacency matrix $\mathbf{W} \in \mathbb{R}^{d \times d}$ (*right*) by learning a directed graph $\mathbf{H} \in \mathbb{R}^{d \times d}$ (*left*) and a priority vector $\mathbf{p} \in \mathbb{R}^d$ on the nodes (*top*). To this end, we introduce a *smooth* acyclic orientation matrix $S_{t,\varepsilon}(\mathbf{p})$ (*center*), a differentiable function of the priority vector where each lower-priority node approximately feeds into each higher-priority node. Gray dashed arrows denote arcs with approximately zero weight. By annealing its temperature during training, the smooth orientation matrix $S_{t,\varepsilon}$ converges to a discrete orientation and, consequently, the overall adjacency matrix $\mathbf{W}$ converges to a DAG.

only requires a quadratic number of operations per optimization step, its constraint-free scheme can be used as a direct and faster replacement for the NOTEARS constrained optimization problem. Overall, we propose a methodology to perform **co**nstraint-free structure learning with **sm**ooth acyclic **o**rientations, which we name COSMO (Figure 1).

**Contributions.** We summarize the key contributions of this paper as follows:

- We introduce a differentiable relaxation of an acyclic orientation matrix, which we name *smooth* orientation matrix (Definition 1). The matrix depends on a temperature value that controls the approximation of the discrete orientation matrix. We prove that we can represent all and only DAGs as the element-wise multiplication of a weighted adjacency matrix and our novel *smooth* orientation matrix (Theorem 1).

- We propose COSMO, an unconstrained optimization approach that learns a DAG entirely avoiding acyclicity constraints (Section 4.2). COSMO represents DAGs through a *smooth* orientation matrix and requires solving a unique optimization problem while annealing the temperature. Since reconstructing the DAG requires a number of operations quadratic on the number of nodes, COSMO is an order of magnitude faster than cubic-expensive constrained methods in literature.

- We connect our proposed scheme to existing constrained approaches and prove that annealing the temperature during training effectively decreases an upper bound on the acyclicity of the *smooth* orientation matrix (Theorem 2).

- We perform a thorough experimental comparison of COSMO and competing structure learning approaches (Section 5). The empirical results report how COSMO achieves comparable structure recovery performances in significantly less time. Further, we highlight how COSMO consistently outperforms previous partially unconstrained structure learning proposals. For reproducibility purposes, we release the necessary code to replicate our experiments.

In the following, we discuss related works in Section 2 and report the necessary background on graph theory and structure learning in Section 3. Then, we introduce COSMO and our original contributions in Section 4. Finally, we report and discuss our empirical analysis in Section 5.

Table 1: Summary comparison of our proposal, COSMO, with competing approaches. We propose a parameterization that enables unconstrained learning of an acyclic graph without trading off on the adjacency matrix rank or the exactness of the acyclicity constraint. To express computational complexity, we define $d$ as the number of nodes and $k$ as the maximum length of iterative approaches. [†]: NOCURL requires a preliminary solution obtained by partially solving a cubic-expensive constrained optimization problem.

| Method | Complexity | Constraint |
|---|---|---|
| NOTEARS (Zheng et al., 2018) | $O(d^3)$ | Exact |
| DAGMA (Bello et al., 2022) | $O(d^3)$ | Exact |
| NOBEARS (Lee et al., 2019) | $O(kd^2)$ | Approximated |
| TMPI (Zhang et al., 2022) | $O(kd^2)$ | Approximated |
| NOCURL (Yu et al., 2021) | $O(d^2)$† | Partial |
| **COSMO** | $\boldsymbol{O(d^2)}$ | **None** |

## 2 RELATED WORKS

Both combinatorial and score-based structure learning approaches must deal with the number of possible DAGs, which grows exponentially with the number of variables (Spirtes & Zhang, 2016). To avoid this issue, a more recent line of research treats the space of DAGs as a continuous space and addresses the structure learning problem as an optimization task (Vowels et al., 2022). In this section, we report related works aiming to improve, approximate, or avoid altogether the constrained optimization scheme firstly introduced by NOTEARS (Zheng et al., 2018), as we summarize in Table 1. Given the shared intuition, we also briefly discuss classical and continuous order learning approaches.

**Low-Rank Approximation.** Several works extended NOTEARS by assuming that the adjacency matrix of the underlying graph does not have full rank either to reduce the number of trainable parameters (Fang et al., 2023) or to improve computational complexity (Lopez et al., 2022). In this work, we deal with possibly full-rank matrices and do not directly compare with low-rank solutions.

**Constraint Reformulation.** NOBEARS (Lee et al., 2019) proposes to estimate the acyclicity constraint by approximating the spectral radius of the adjacency matrix. Given a maximum number $k$ of iterations, the constraint can then be evaluated on a graph with $d$ nodes in $O(kd^2)$ time. Similarly, TMPI (Zhang et al., 2022) proposes an iterative approximation of the constraint that also results in $O(kd^2)$ computational complexity. Finally, DAGMA (Bello et al., 2022) reformulates the acyclicity constraint as the log-determinant of a linear transformation of the adjacency matrix. While still asymptotically cubic in complexity, the use of log-determinant is significantly faster in practice because of widespread optimizations in common linear algebra libraries (Bello et al., 2022, pp.19).

**Unconstrained Methods.** To tackle the exponential size of the space of DAGs, several works follow the intuition of separately fitting the graph orientation and its adjacencies (Friedman & Koller, 2003; Teyssier & Koller, 2012; Bernstein et al., 2020; Deng et al., 2023). In the context of differentiable approaches, of particular interest are causal discovery methods that avoid acyclicity constraints by fitting a distribution over permutations parameterized either explicitly (Cundy et al., 2021; Charpentier et al., 2022) or through their topological ordering (Zantedeschi et al., 2022). While sharing the overall intuition of fitting DAGs through their orientation, we propose a parameterization and an optimization scheme to fastly optimize acyclic adjacencies that could be then easily integrated in causal discovery methods. For this reason, we focus our analysis on competing acyclic optimization methods. However, we discuss the comparison with a more complete approach, namely DAGUERREOTYPE (Zantedeschi et al., 2022), in Appendix E.19. Always in the context of causal discovery, ENCO (Lippe et al., 2022) decouples a DAG in an adjacency matrix and an edge orientation matrix. The authors explicitly parameterize the orientation matrix and prove that it converges to an acyclic orientation whenever the training dataset contains a sufficient number of interventions. Our structure learning proposal tackles instead non-intervened datasets and ensures acyclicity by construction. Similarly to us, NOCURL (Yu et al., 2021) proposes a model that decouples the topological ordering from the adjacencies of an

acyclic graph. However, the proposed optimization schemes are significantly different. Firstly, their approach extracts the nodes ordering from a preliminary solution obtained by partially solving the NOTEARS constrained optimization problem. Then, they fix the ordering and unconstrainedly optimize only the direct adjacency matrix. On the other hand, COSMO jointly learns priorities and adjacencies avoiding entirely acyclicity evaluations. Finally, GOLEM proposes an unconstrained optimization problem regularized with the NOTEARS acyclicity constraint, which still needs to be evaluated on each optimization step and might nonetheless lead to cyclic DAGs (Ng et al., a). We report further discussion on the theoretical comparison with ENCO and NOCURL in Appendix D and we carefully empirically compare with NOCURL in Section 5.

## 3 BACKGROUND

**Graph Theory.** A *directed graph* is a pair $D = (V, A)$ of vertices $V = \{1, \ldots, d\}$ and arcs between them $A \subseteq V \times V$. A directed *acyclic* graph (DAG) is a directed graph whose arcs follow a strict partial order on the vertices. In a DAG, the *parents* of a vertex $v \in V$ are the set of incoming nodes such that $\mathrm{pa}(v) = \{u \in V \mid (u, v) \in A\}$ (Bondy & Murty, 2008). We represent a directed graph as a binary adjacency matrix $\mathbf{A} \in \{0, 1\}^{d \times d}$, where $\mathbf{A}_{uv} \neq 0 \iff (u, v) \in A$. Similarly, we define a weighted adjacency matrix as the real matrix $\mathbf{W} \in \mathbb{R}^{d \times d}$, where $\mathbf{W}_{uv} \neq 0 \iff (u, v) \in A$.

**Structure Learning.** A Structural Equation Model (SEM) models a data-generating process as a set of functions $f = \{f_1, \ldots, f_d\}$, where $f_v : \mathbb{R}^{|\mathrm{pa}(v)|} \to \mathbb{R}$ for each variable $v \in V$ in the DAG (Pearl, 2009). Given a class of functions $\mathbf{F}$ and a loss $\mathcal{L}$, NOTEARS (Zheng et al., 2020) formalizes non-linear acyclic structure learning through the following constrained optimization problem

$$\min_{f \in \mathbf{F}} \mathcal{L}(f) \quad \text{s.t.} \quad \mathrm{tr}(e^{W(f) \circ W(f)}) - d = 0, \tag{1}$$

where $W(f) \in \mathbb{R}^{d \times d}$ is the adjacency matrix representing parent relations between variables in $f$. In particular, the constraint equals zero if and only if the adjacency matrix $W(f)$ is acyclic. The authors propose to solve the problem using the Augmented Lagrangian method (Nocedal & Wright, 1999), which in turn requires to solve multiple unconstrained problems and to compute the constraint value at each optimization step. Notably, any causal interpretation of the identified arcs depends on several assumptions on both the function class $\mathbf{F}$ and the loss function $\mathcal{L}$ (van de Geer & Bühlmann, 2013; Loh & Bühlmann, 2014).

## 4 LEARNING ACYCLIC ORIENTATIONS WITH COSMO

In Subsection 4.1, we propose to parameterize a weighted adjacency matrix as a function of a direct matrix and a smooth orientation matrix. In this way, we effectively express the discrete space of DAGs in a continuous and differentiable manner. Then, in Subsection 4.2, we introduce COSMO, an unconstrained optimization approach to learn acyclic DAGs. Furthermore, we prove an upper bound on the acyclicity of the smooth orientation matrix that connects our formulation to constrained approaches. To ease the presentation, we initially assume linear relations between variables. By doing so, the weighted adjacency matrix is the unique parameter of the problem. However, as with previous structure learning approaches, our proposal easily extends to non-linear models by jointly optimizing a non-linear model and an adjacency matrix either weighting or masking variables dependencies. We report one possible extension of COSMO to non-linear relations in Appendix C.3.

### 4.1 SMOOTH ACYCLIC ORIENTATIONS

To continuously represent the space of DAGs with $d = |V|$ nodes, we introduce a priority vector $\mathbf{p} \in \mathbb{R}^d$ on its vertices. Consequently, given the priority vector $\mathbf{p}$ and a strictly positive threshold $\varepsilon > 0$, we define the following strict partial order $\prec_{\mathbf{p}, \varepsilon}$ on the vertex set $V$

$$\forall (u, v) \in V \times V : \quad u \prec_{\mathbf{p}, \varepsilon} v \iff \mathbf{p}_v - \mathbf{p}_u \geq \varepsilon. \tag{2}$$

In other terms, a vertex $u$ precedes another vertex $v$ if and only if the priority of $v$ is sufficiently larger than the priority of the vertex $u$. Notably, with a zero threshold $\varepsilon = 0$, the relation would be symmetric and thus not a strict order. On the other hand, whenever $\varepsilon$ is strictly positive, we can represent a subset of all strict partial orders sufficient to express all possible DAGs.

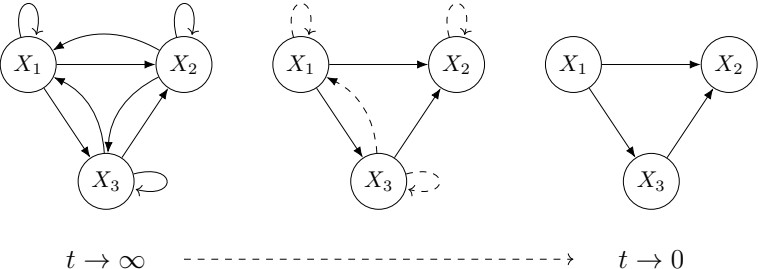

Figure 2: (*left*) With infinite temperature, the sigmoid function is constant and connects all vertices. (*center*) Given two nodes, for positive temperatures the smooth orientation matrix has larger values on the arcs respecting the priority ordering. (*right*) In the limit of the temperature to zero, the smooth orientation matrix contains non-zero entries if and only if the arc respects the order, i.e., it directs a node to another with sufficiently higher priority.

**Lemma 1.** *Let* $\mathbf{W} \in \mathbb{R}^{d \times d}$ *be a real matrix. Then, for any* $\varepsilon > 0$, $\mathbf{W}$ *is the weighted adjacency matrix of a DAG if and only if it exists a priority vector* $\mathbf{p} \in \mathbb{R}^d$ *and a real matrix* $\mathbf{H} \in \mathbb{R}^{d \times d}$ *such that*

$$\mathbf{W} = \mathbf{H} \circ \mathbf{T}_{\prec_{\mathbf{p}, \varepsilon}}, \tag{3}$$

*where* $\mathbf{T}_{\prec_{\mathbf{p}, \varepsilon}} \in \{0, 1\}^{d \times d}$ *is a binary orientation matrix such that*

$$\mathbf{T}_{\prec_{\mathbf{p}, \varepsilon}}[uv] = \begin{cases} 1 & \text{if } u \prec_{\mathbf{p}, \varepsilon} v \\ 0 & \text{otherwise,} \end{cases} \tag{4}$$

*for any* $u, v \in V$.

*Proof.* We report the proof in Appendix A.1. $\qquad\square$

While priority vectors enable the representation of strict partial orders in a continuous space, the construction of the orientation matrix still requires the non-differentiable evaluation of the inequality between priority differences from Equation 2. To this end, we approximate the comparison of the difference against the threshold $\varepsilon$, using a *tempered* sigmoidal function. We refer to such approximation of the orientation matrix as the *smooth* orientation matrix.

**Definition 1** (Smooth Orientation Matrix). *Let* $\mathbf{p} \in \mathbb{R}^d$ *be a priority vector,* $\varepsilon > 0$ *be a strictly positive threshold, and* $t > 0$ *be a strictly positive temperature. Then, the* smooth *orientation matrix of the strict partial order* $\prec_{\mathbf{p}, \varepsilon}$ *is the real matrix* $S_{t, \varepsilon}(\mathbf{p}) \in \mathbb{R}^{d \times d}$ *such that, for any* $u, v \in V$, *it holds*

$$S_{t, \varepsilon}(\mathbf{p})_{uv} = \sigma_{t, \varepsilon}(\mathbf{p}_v - \mathbf{p}_u), \tag{5}$$

*where* $\sigma_{t, \varepsilon}$ *is the* $\varepsilon$-centered tempered sigmoid, defined as

$$\sigma_{t, \varepsilon}(x) = \frac{1}{1 + e^{-(x - \varepsilon)/t}}. \tag{6}$$

Intuitively, the threshold $\varepsilon$ shifts the center of the sigmoid and breaks the symmetry whenever two variables *approximately* have the same priority. The temperature parameter $t > 0$ regulates instead the steepness of the sigmoid. Because of the asymmetry introduced by the threshold, in the limit of the temperature to zero, the zero-entries of a smooth orientation matrix coincide with the zero-entries of the corresponding orientation matrix (Figure 2). Therefore, we prove that any directed acyclic graph can be represented as the element-wise product of a directed adjacency matrix and a smooth orientation. Further, any directed graph resulting from this decomposition is acyclic.

**Theorem 1.** *Let* $\mathbf{W} \in \mathbb{R}^{d \times d}$ *be a real matrix. Then, for any* $\varepsilon > 0$, $\mathbf{W}$ *is the weighted adjacency matrix of a DAG if and only if it exists a priority vector* $\mathbf{p} \in \mathbb{R}^d$ *and a real matrix* $\mathbf{H} \in \mathbb{R}^{d \times d}$ *such that*

$$\mathbf{W} = \mathbf{H} \circ \lim_{t \to 0} S_{t, \varepsilon}(\mathbf{p}), \tag{7}$$

*where* $S_{t, \varepsilon}(\mathbf{p})$ *is the smooth orientation matrix of* $\prec_{\mathbf{p}, \varepsilon}$.

*Proof.* We report the proof in Appendix A.2. $\qquad\square$

## 4.2 LEARNING ADJACENCIES AND ORIENTATIONS

Given our definition of smooth acyclic orientation (Definition 1), we can effectively parameterize the space of DAGs as a continuous function of a weighted adjacency matrix $\mathbf{H} \in \mathbb{R}^{d \times d}$ and a priority vector $\mathbf{p} \in \mathbb{R}^d$. Therefore, the computational complexity of our solution reduces to the construction of the adjacency matrix $\mathbf{W} = \mathbf{H} \circ S_{t,\varepsilon}(\mathbf{p})$, which can be achieved in $O(d^2)$ time and space per optimization step by computing each arc as $\mathbf{W}_{uv} = \mathbf{H}_{uv} \cdot \sigma((\mathbf{p}_v - \mathbf{p}_u - \varepsilon)/t)$. In the literature, NOCURL proposed a similar model where each arc has form $\mathbf{W}_{uv} = \mathbf{H}_{uv} \cdot \text{ReLU}(\mathbf{p}_v - \mathbf{p}_u)$. Despite the similarity, NOCURL is trained with a significantly different optimization scheme, as we also discuss in Appendix D.2. In fact, to avoid a significant performance drop, their formulation requires a preliminary solution from a constrained optimization problem and does not jointly learn the parameters corresponding to our adjacencies and priorities. In the following, we describe how COSMO effectively reduces to an unconstrained problem and avoids evaluating acyclicity altogether.

**Temperature Annealing.** The smooth orientation matrix $S_{t,\varepsilon}(\mathbf{p})$ represents an acyclic orientation only in the limit of the temperature $t \to 0$. Nonetheless, the gradient loss vanishes whenever the temperature tends to zero. In fact, for an arbitrary loss function $\mathcal{L}$, we can decompose the gradient of each component $\mathbf{p}_u$ of the priority vector as follows

$$\frac{\partial \mathcal{L}(\mathbf{W})}{\partial \mathbf{p}_u} = \sum_{v \in V} \frac{\partial \mathcal{L}(\mathbf{W})}{\partial \mathbf{W}_{uv}} \cdot \frac{\partial \mathbf{W}_{uv}}{\partial \mathbf{p}_u} + \frac{\partial \mathcal{L}(\mathbf{W})}{\partial \mathbf{W}_{vu}} \cdot \frac{\partial \mathbf{W}_{vu}}{\partial \mathbf{p}_u}, \tag{8}$$

$$\frac{\partial \mathbf{W}_{uv}}{\partial \mathbf{p}_u} = -\frac{\mathbf{H}_{uv}}{t} \sigma_{t,\varepsilon}(\mathbf{p}_v - \mathbf{p}_u)(1 - \sigma_{t,\varepsilon}(\mathbf{p}_v - \mathbf{p}_u)) \tag{9}$$

$$\frac{\partial \mathbf{W}_{vu}}{\partial \mathbf{p}_u} = \frac{\mathbf{H}_{vu}}{t} \sigma_{t,\varepsilon}(\mathbf{p}_v - \mathbf{p}_u)(1 - \sigma_{t,\varepsilon}(\mathbf{p}_v - \mathbf{p}_u)). \tag{10}$$

Therefore, by property of the sigmoidal function $\sigma_{t,\varepsilon}$ it holds that both $\partial \mathbf{W}_{vu}/\partial \mathbf{p}_u$ and $\partial \mathbf{W}_{vu}/\partial \mathbf{p}_u$ tend to zero for $t \to 0$. To handle this issue, we tackle the optimization problem by progressively reducing the temperature during training. In practice, we perform cosine annealing from an initial positive value $t_{\text{start}}$ to a significantly lower target value $t_{\text{end}} \approx 0$. We further motivate our choice by showing the existence of an upper bound on the acyclicity of the orientation matrix that is a monotone increasing function of the temperature. Therefore, temperature annealing effectively decreases the acyclicity upper bound during training of the *smooth* orientation and, consequently, of the adjacencies.

**Theorem 2.** *Let $\mathbf{p} \in \mathbb{R}^d$ be a priority vector, $\varepsilon > 0$ be a strictly positive threshold, and $t > 0$ be a strictly positive temperature. Then, given the smooth orientation matrix $S_{t,\varepsilon}(\mathbf{p}) \in \mathbb{R}^{d \times d}$, it holds*

$$h(S_{t,\varepsilon}(\mathbf{p})) \leq e^{d\alpha} - 1, \tag{11}$$

*where $h(S_{t,\varepsilon}(\mathbf{p})) = \text{tr}(e^{S_{t,\varepsilon}(\mathbf{p})}) - d$ is the NOTEARS acyclicity constraint and $\alpha = \sigma(-\varepsilon/t)$.*

*Proof.* We report the proof in Appendix B. $\square$

**Direct Matrix Regularization.** To contrast the discovery of spurious arcs we perform feature selection by applying L1 regularization on the adjacency matrix $\mathbf{H}$. Further, during the annealing procedure, even if a vertex $u$ precedes $v$ in the partial order $\prec_{\mathbf{p},\varepsilon}$, the weight of the opposite arc $v \to u$ in the smooth orientation matrix will only be approximately zero. Therefore, sufficiently large values of the weighted adjacency matrix $\mathbf{H}$, might still lead to undesirable cyclic paths during training. To avoid this issue, we regularize the L2-norm of the non-oriented adjacency matrix.

**Priority Vector Regularization.** Other than for small temperature values, the partial derivatives in Equations 9 and 10 tend to zero whenever the priorities distances $|\mathbf{p}_v - \mathbf{p}_u|$ tend to infinity. Therefore, we regularize the L2-norm of the priority vector. For the same reason, we initialize each component from the normal distribution $\mathbf{p}_u \sim \mathcal{N}(0, \varepsilon^2/2)$, so that each difference follows the normal distribution $\mathbf{p}_v - \mathbf{p}_u \sim \mathcal{N}(0, \varepsilon^2)$. We provide further details on initialization in Appendix A.3.

**Optimization Problem.** We formalize COSMO as the differentiable and unconstrained problem

$$\min_{\mathbf{H} \in \mathbb{R}^{d \times d}, \mathbf{p} \in \mathbb{R}^d} \mathcal{L}(\mathbf{H} \circ S_{t,\varepsilon}(\mathbf{p})) + \lambda_1 \|\mathbf{H}\|_1 + \lambda_2 \|\mathbf{H}\|_2 + \lambda_p \|\mathbf{p}\|_2, \tag{12}$$

where $\lambda_1, \lambda_2, \lambda_p$ are the regularization coefficients for the adjacencies and the priorities. As the regularization coefficients $\boldsymbol{\lambda} = \{\lambda_1, \lambda_2, \lambda_p\}$, the initial temperature $t_{\text{start}}$, the target temperature $t_{\text{end}}$, and the shift $\varepsilon$ are hyperparameters of our proposal, whose choice we discuss in Appendix C.4. Notably, Theorem 2 can also guide the choice of the final temperature value and the shift to guarantee a maximum tolerance on the acyclicity of the smooth orientation matrix.

## 5  EXPERIMENTS

We present an experimental comparison of COSMO against related acyclic structure learning approaches. Our method operates on possibly full-rank graphs and ensures the exact acyclicity of the solution. Therefore, we focus on algorithms providing the same guarantees and under the same conditions. Namely, we confront with the structure learning performance and execution time of NOTEARS (Zheng et al., 2018), NOCURL (Yu et al., 2021), and DAGMA (Bello et al., 2022). As previously discussed, NOCURL proposes a similar model with a substantially different optimization scheme. To highlight the importance of both our parameterization and optimization scheme, we also compare with an entirely unconstrained variant of the algorithm where we directly train the variables ordering without any preliminary solution. In the results, we refer to this variant as NOCURL-U.

We base our empirical analysis on the testbed originally introduced by Zheng et al. (2018) and then adopted as a benchmark by all followup methods. In particular, we test continuous approaches on randomly generated Erdös-Rényi (ER) and scale-free (SF) graphs of increasing size and for different exogenous noise types. For each method, we perform structure learning by minimizing the Mean Squared Error (MSE) of a model on a synthetic dataset using the Adam optimizer (Kingma & Ba, 2015). In Appendix C, we report further details on the implementation of COSMO, the baselines, and the datasets. We include the code to reproduce our experiments in the Supplementary Materials.[1]

### 5.1  EVALUATION OVERVIEW

In line with previous work, we retrieve the binary adjacency matrix by thresholding the learned weights against a small threshold $\omega = 0.3$ (Zheng et al., 2018). While COSMO guarantees the solution to be acyclic, we maintain the thresholding step to prune correctly oriented but spurious arcs. Then, we measure the Normalized Hamming Distance (NHD) between the solution and the ground-truth as the sum of missing, extra, or incorrect edges divided by the number of nodes. In general, testing weights against a fixed threshold might limit the retrieval of significant arcs with small coefficients in the true model (Xu et al., 2022). For this reason, we also compute the Area under the ROC curve (AUC), which describes the trade-off between the True Positive Rate (TPR) and the False Positive Rate (FPR) for increasing values of the weight threshold (Heinze-Deml et al., 2018). Due to space limitations, we only report in the main body the AUC results, which is the most comprehensive score. We provide detailed results for other metrics, including NHD, in Appendix E.

### 5.2  RESULTS DISCUSSION

By looking at the AUC of the learned graphs, we observe that COSMO consistently achieves results that are comparable and competitive with those from constrained-optimization solutions such as DAGMA or NOTEARS across different graph sizes and noise types (Table 2). This empirically confirms the approximation properties of COSMO, which can reliably discover DAGs without resorting to explicit acyclicity constraints.

Furthermore, COSMO performs better than NOCURL on most datasets. We recall that the latter is the only existing structure learning approach combining constrained and unconstrained optimization. As pointed out in Yu et al. (2021), we also observe that the discovery performance of NOCURL drops when optimizing the variable ordering instead of inferring it from a preliminary solution. The fact that COSMO outperforms NOCURL-U on all datasets highlights the substantial role and effect of our *smooth* orientation formulation and our optimization scheme for learning the topological ordering of variables from data in an unconstrained way. Overall, our proposal achieves, on average, the best or the second-best result for the AUC metric across all the analyzed datasets and correctly classifies arcs also for large graphs (Figure 3). Further, as we extensively report in Appendix E for different graph

---

[1]https://github.com/rmassidda/cosmo

Table 2: Experimental results on linear ER-4 acyclic graphs with different noise terms and sizes. For each algorithm, we report mean and standard deviation over five independent runs of the AUC metric and the time in seconds. We highlight in bold the **best** result and in italic bold the *second best* result. The reported duration of NOCURL includes the time to retrieve the necessary preliminary solution using an acyclicity constraint. We denote as NOCURL-U the quadratic version of NOCURL. Complete results on additional metrics and graph types are in Appendix E.

| $d$ | Algorithm | Gauss | | Exp | | Gumbel | |
|---|---|---|---|---|---|---|---|
| | | AUC | Time | AUC | Time | AUC | Time |
| 30 | COSMO | *0.984 ± 0.02* | **88 ± 2** | **0.989 ± 0.01** | **89 ± 3** | 0.914 ± 0.10 | **87 ± 2** |
| | DAGMA | **0.985 ± 0.01** | 781 ± 192 | *0.986 ± 0.02* | 744 ± 75 | *0.973 ± 0.02* | 787 ± 86 |
| | NOCURL | 0.967 ± 0.01 | 822 ± 15 | 0.956 ± 0.02 | 826 ± 24 | 0.915 ± 0.04 | 826 ± 17 |
| | NOCURL-U | 0.694 ± 0.06 | *226 ± 5* | 0.694 ± 0.05 | *212 ± 5* | 0.678 ± 0.05 | *212 ± 5* |
| | NOTEARS | 0.973 ± 0.02 | 5193 ± 170 | 0.966 ± 0.03 | 5579 ± 284 | **0.981 ± 0.01** | 5229 ± 338 |
| 100 | COSMO | 0.961 ± 0.03 | **99 ± 2** | *0.985 ± 0.01* | **99 ± 2** | *0.973 ± 0.01* | **98 ± 1** |
| | DAGMA | **0.982 ± 0.01** | 660 ± 141 | **0.986 ± 0.01** | 733 ± 109 | **0.986 ± 0.01** | 858 ± 101 |
| | NOCURL | 0.962 ± 0.01 | 1664 ± 14 | 0.950 ± 0.02 | 1655 ± 28 | 0.962 ± 0.01 | 1675 ± 34 |
| | NOCURL-U | 0.682 ± 0.05 | *267 ± 10* | 0.693 ± 0.05 | *242 ± 4* | 0.663 ± 0.04 | *247 ± 9* |
| | NOTEARS | *0.963 ± 0.01* | 11000 ± 339 | 0.972 ± 0.01 | 10880 ± 366 | 0.969 ± 0.00 | 11889 ± 343 |
| 500 | COSMO | *0.933 ± 0.01* | **436 ± 81** | **0.986 ± 0.00** | **390 ± 102** | **0.982 ± 0.01** | **410 ± 106** |
| | DAGMA | **0.980 ± 0.00** | 2485 ± 365 | *0.984 ± 0.01* | 2575 ± 469 | *0.980 ± 0.00* | 2853 ± 218 |
| | NOCURL-U | 0.683 ± 0.05 | *1546 ± 304* | 0.715 ± 0.03 | *1488 ± 249* | 0.728 ± 0.05 | *1342 ± 209* |

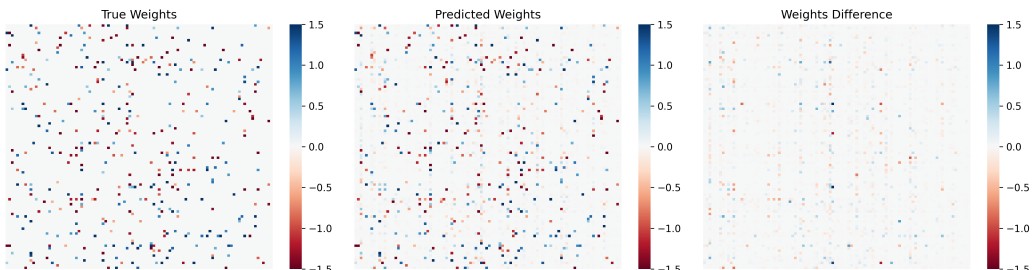

Figure 3: Visualization of the weighted adjacency matrix learned by COSMO (ER4, Gaussian noise, 100 nodes) against the ground truth. We also report the difference between the ground-truth and the learned weights. By thresholding the learned weighted adjacency matrix, COSMO correctly classifies most true (TPR = 0.96) and non-existing arcs (FPR = 0.01), resulting in a limited number of errors (NHD = 0.93) due to the narrow difference in the retrieved weights.

size and noise terms, our non-linear extension obtains comparable performances with DAGMA-MLP in significantly less time.

Unsurprisingly, due to its quadratic computational complexity, COSMO is significantly faster than constrained methods on all datasets, especially for increasing graph sizes. Notably, despite employing early stopping conditions for convergence, all competing methods incur in the cost of solving multiple optimization problems with higher computational cost per step (Figure 4). In particular, while the unconstrained variant NOCURL-U has a comparable per-epoch average time cost, for a substantially worse graph recovery performance, NOCURL overall pays the need for a preliminary solution computed with an acyclicity constraint. Therefore, already for graphs with 500 nodes, only COSMO, DAGMA, and NOCURL-U return a solution before hitting our wall time limit. Finally, we observe that the cubic computational complexity of DAGMA significantly emerges when dealing with large graphs. Therefore, despite the effective underlying optimizations on the log-determinant computation, DAGMA's acyclicity constraint still affects scalability.

Given the proposed parameterization, COSMO requires particular care in the choice of the regularization hyperparameters. In particular, we carefully considered the importance of regularizing the priority vector, which constitutes one of our main differences with previous structure learning approaches.

Table 3: Ablation test of the priority regularization term $\lambda_p$ on DAGs with different noise terms and sizes. We name the configuration without priority regularization as COSMO-NP and report mean and standard deviation over five independent runs. For each configuration, the **best** result is in bold.

| Graph | $d$ | Algorithm | NHD | TPR | AUC |
|---|---|---|---|---|---|
| ER4 | 30 | COSMO | $\mathbf{0.867 \pm 1.01}$ | $\mathbf{0.953 \pm 0.04}$ | $\mathbf{0.984 \pm 0.02}$ |
| | | COSMO-NP | $1.893 \pm 0.92$ | $0.870 \pm 0.07$ | $0.937 \pm 0.05$ |
| | 100 | COSMO | $\mathbf{1.388 \pm 0.69}$ | $0.917 \pm 0.04$ | $0.961 \pm 0.03$ |
| | | COSMO-NP | $1.570 \pm 0.56$ | $\mathbf{0.935 \pm 0.04}$ | $\mathbf{0.974 \pm 0.02}$ |
| ER6 | 30 | COSMO | $\mathbf{4.087 \pm 1.12}$ | $\mathbf{0.838 \pm 0.06}$ | $\mathbf{0.921 \pm 0.04}$ |
| | | COSMO-NP | $4.153 \pm 2.38$ | $0.819 \pm 0.12$ | $0.885 \pm 0.09$ |
| | 100 | COSMO | $\mathbf{9.476 \pm 3.01}$ | $0.771 \pm 0.08$ | $0.911 \pm 0.05$ |
| | | COSMO-NP | $9.804 \pm 2.90$ | $\mathbf{0.848 \pm 0.05}$ | $\mathbf{0.941 \pm 0.03}$ |

We found that our hyperparameter search procedure consistently returned relatively low priority regularization values ($\lambda_p \approx$ 1e-3). However, while it might benefit structure learning for larger graphs, ablating priority regularization results in a non-negligible performance drop for smaller graphs (Table 3).

Another important aspect of our proposal is the temperature annealing procedure. While we proved that in the limit of the temperature to zero the resulting model will be acyclic, the same procedure might lead to zero-gradients effectively impeding the optimization. In Appendix G, we show that COSMO consistently reaches acyclic solutions before completely annealing the temperature and thus that optimization continues even in the last epochs. Finally, we also find empirical evidence that our L2 regularization scheme on the directed adjacency matrix effectively contains acyclicity during training (Appendix F).

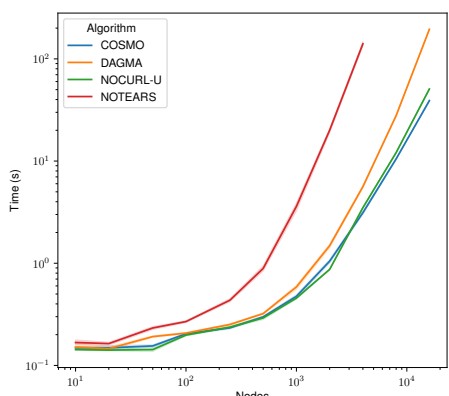

Figure 4: Average duration of a training epoch for an increasing number of nodes on five independent runs on random ER-4 DAGs.

## 6 CONCLUSION

We introduced COSMO, an unconstrained and continuous approach for recovering DAGs from data. Our novel definition of *smooth* orientation matrix ensures the acyclicity of the solution without requiring the evaluation of computationally expensive constraints. Furthermore, we prove that annealing the temperature of our smooth acyclic orientation corresponds to decreasing an upper bound on the widely adopted acyclicity relaxation from NOTEARS. Overall, our empirical analysis showed that COSMO performs comparably to constrained methods in significantly less time. Notably, our proposal significantly outperforms the only existing work *partially* optimizing in the space of DAGs, NOCURL, and its completely unconstrained variant NOCURL-U. Experimental results highlight the role of our parameterization, which does not incur the necessity of preliminary solutions and provably returns a DAG without ever evaluating acyclicity.

In recent years, several authors debated using continuous acyclic learners as full-fledged causal discovery algorithms (Reisach et al., 2021; Kaiser & Sipos, 2022; Ng et al., b). In this context, our empirical analysis of COSMO shares the same limitations of existing baselines and, exactly like them, might not be significant in the causal discovery scenario. However, acyclic optimization techniques are a fundamental component of continuous discovery approaches (Brouillard et al., 2020; Lorch et al., 2022). By reducing by an order of magnitude the necessary time to optimize an acyclic causal graph, COSMO opens up more scalable continuous causal discovery strategies without sacrificing — as demonstrated in this work — the theoretical guarantees on DAGs approximation capabilities.

ACKNOWLEDGMENTS

The work has been partially supported by the EU H2020 TAILOR project (n.952215).

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

## A  DEFERRED PROOFS

### A.1  PROOF OF LEMMA 1

**Lemma 1**  *Let* $\mathbf{W} \in \mathbb{R}^{d \times d}$ *be a real matrix. Then, for any* $\varepsilon > 0$, $\mathbf{W}$ *is the weighted adjacency matrix of a DAG if and only if it exists a priority vector* $\mathbf{p} \in \mathbb{R}^d$ *and a real matrix* $\mathbf{H} \in \mathbb{R}^{d \times d}$ *such that*

$$\mathbf{W} = \mathbf{H} \circ \mathbf{T}_{\prec_{\mathbf{p},\varepsilon}}, \tag{13}$$

*where* $\mathbf{T}_{\prec_{\mathbf{p},\varepsilon}} \in \{0,1\}^{d \times d}$ *is a binary orientation matrix such that*

$$\mathbf{T}_{\prec_{\mathbf{p},\varepsilon}}[uv] = \begin{cases} 1 & \text{if } u \prec_{\mathbf{p},\varepsilon} v \\ 0 & \text{otherwise,} \end{cases} \tag{14}$$

*for any* $u, v \in V$.

*Proof.* Firstly, we prove the existence of a priority vector $\mathbf{p}$ and an adjacency matrix $\mathbf{H}$ for each weighted acyclic matrix $\mathbf{W}$ of a directed acyclic graph $D = (V, A)$. Being a DAG, the arcs follow a strict partial order $\prec$ on the vertices $V = \{1, \ldots, d\}$. Therefore, it holds that

$$A \subseteq \{(u, v) \mid u \prec v\}. \tag{15}$$

Consequently, for an arbitrary topological ordering of the variables $\pi \colon V \to \{1, \ldots, d\}$, which always exists on DAGs, we define the vector $\mathbf{p} \in \mathbb{R}^d$ such that

$$\mathbf{p}_u = \varepsilon \pi(u). \tag{16}$$

Given the following implications

$$u \prec v \implies \pi(v) > \pi(u) \tag{17}$$

$$\implies \mathbf{p}_v - \mathbf{p}_u = \varepsilon(\pi(v) - \pi(u)) \geq \varepsilon \tag{18}$$

$$\iff u \prec_{\mathbf{p},\varepsilon} v, \tag{19}$$

it holds that the order $\prec_{\mathbf{p},\varepsilon}$ contains the order $\prec$. Finally, we can define the adjacency matrix as $\mathbf{H} = \mathbf{W}$, where $\mathbf{W} = \mathbf{H} \circ \mathbf{T}_{\prec_{\mathbf{p},\varepsilon}}$ holds since $\mathbf{T}_{\prec_{\mathbf{p},\varepsilon}}[u,v] = 0$ only if $(u,v) \notin A$.

To prove that any priority vector $\mathbf{p}$ and adjacency matrix $\mathbf{H}$ represent a DAG, we first notice that, since the arcs follow a strict partial order, the orientation $\mathbf{T}_{\prec_{\mathbf{p},\varepsilon}}$ is acyclic. Then, by element-wise multiplying any matrix $\mathbf{H}$ we obtain a sub-graph of a DAG, which is acyclic by definition. $\square$

### A.2  PROOF OF THEOREM 1

**Theorem 1**  *Let* $\mathbf{W} \in \mathbb{R}^{d \times d}$ *be a real matrix. Then, for any* $\varepsilon > 0$, $\mathbf{W}$ *is the weighted adjacency matrix of a DAG if and only if it exists a priority vector* $\mathbf{p} \in \mathbb{R}^d$ *and a real matrix* $\mathbf{H} \in \mathbb{R}^{d \times d}$ *such that*

$$\mathbf{W} = \mathbf{H} \circ \lim_{t \to 0} S_{t,\varepsilon}(\mathbf{p}), \tag{20}$$

*where* $S_{t,\varepsilon}(\mathbf{p})$ *is the smooth orientation matrix of* $\prec_{\mathbf{p},\varepsilon}$.

*Proof.* By Lemma 1, we know that for any acyclic weighted adjacency matrix $\mathbf{W}$ there exist a priority vector $\mathbf{p}$ and a real matrix $\mathbf{H}$ such that $\mathbf{W} = \mathbf{H} \circ \mathbf{T}_{\prec_{\mathbf{p},\varepsilon}}$. Further, by Definition 1, the inner limit of Equation 20 solves to

$$\lim_{t \to 0} S_{t,\varepsilon}(\mathbf{p})_{uv} = \begin{cases} 1 & \mathbf{p}_v - \mathbf{p}_u > \varepsilon \\ 1/2 & \mathbf{p}_v - \mathbf{p}_u = \varepsilon \\ 0 & \mathbf{p}_v - \mathbf{p}_u < \varepsilon. \end{cases} \tag{21}$$

Therefore, we can define $\mathbf{H}' \in \mathbb{R}^{d \times d}$ such that

$$\mathbf{H}_{uv} = \begin{cases} 2\mathbf{H}'_{uv} & \mathbf{p}_v - \mathbf{p}_u = \varepsilon, \\ \mathbf{H}'_{uv} & \text{otherwise.} \end{cases} \tag{22}$$

from which

$$\mathbf{W} = \mathbf{H} \circ \mathbf{T}_{\prec_{\mathbf{p},\varepsilon}} = \mathbf{H}' \circ \lim_{t \to 0} \mathbf{S}_{t,\varepsilon}(\mathbf{p}). \tag{23}$$

Then, to prove the counter-implication of Theorem 1, we notice that

$$\lim_{t \to 0} \mathbf{S}_{t,\varepsilon}(\mathbf{p})_{uv} = 0 \iff \mathbf{p}_v - \mathbf{p}_u < \varepsilon \iff u \not\prec_{\mathbf{p},\varepsilon} v. \tag{24}$$

Therefore, since the smooth orientation contains an arc if and only if the vertex respect the strict partial order $\prec_{\mathbf{p},\varepsilon}$, it is acyclic. Consequently, as in Lemma 1, the element-wise product with an acyclic matrix results in a sub-graph of a DAG, which is also acyclic by definition. □

### A.3 PRIORITY VECTOR INITIALIZATION

By independently sampling each priority component from a Normal distribution $\mathcal{N}(\mu, s^2/2)$, each difference is consequently sampled from the distribution $\mathcal{N}(0, s^2)$. Therefore, we seek a value for the standard deviation $s$ that maximizes the partial derivative

$$\frac{\partial \mathbf{W}_{uv}}{\partial \mathbf{p}_u} = \frac{\mathbf{H}_{uv}}{t} \sigma_{t,\varepsilon}(\mathbf{p}_v - \mathbf{p}_u)(1 - \sigma_{t,\varepsilon}(\mathbf{p}_v - \mathbf{p}_u)). \tag{25}$$

for arbitrary vertices $u, v$. Given the definition of the tempered-shifted sigmoid function, this object has maximum in $\mathbf{p}_v - \mathbf{p}_u = \varepsilon$. Therefore, by setting the variance as $s^2 = \varepsilon^2$, we maximize the density function of the point $\mathbf{p}_v - \mathbf{p}_u = \varepsilon$ in the distribution $\mathcal{N}(0, s^2)$.

## B SMOOTH ACYCLIC ORIENTATIONS AND THE ACYCLICITY CONSTRAINT

In this section, we present the proof for the upper bound on the acyclicity of a smooth acyclic orientation matrix. To this end, we introduce two auxiliary and novel lemmas. Firstly, we introduce a lemma which binds the product of a sigmoid on a sequence of values with zero sum (Lemma 2). Then, we introduce another lemma on the sum of the priority differences in a cyclic path (Lemma 3). Finally, we are able to prove the acyclicity upper bound from Theorem 2.

**Lemma 2.** *(Sigmoid Product Upper Bound) Let $\{x_i\}$ be a sequence of $n$ real numbers such that*

$$\sum_{i=1}^{n} x_i = 0.$$

*Then, for any temperature $t > 0$ and shift $\varepsilon \geq 0$, it holds that*

$$\prod_{i=1}^{n} \sigma_{t,\varepsilon}(x_i) \leq \alpha^n,$$

*where $\alpha = \sigma(-\varepsilon/t)$ is the value of the tempered and shifted sigmoid in zero.*

*Proof.* Before starting, we invite the reader to notice that, for any temperature $t > 0$, if the sum of the sequence $\{x_i\}$ is zero, then also the sequence $\{x_i/t\}$ sums to zero. Therefore, we omit the temperature in the following proof, and assume to divide beforehand all elements of the sequence by the temperature $t$. Further, we explicitly denote the shifted sigmoid by using the notation $\sigma(x_i - \varepsilon)$.

Firstly, we formulate the left-side of the inequality as

$$\prod_{i=1}^{n} \sigma(x_i - \varepsilon) = \prod_{i=1}^{n} \frac{e^{x_i - \varepsilon}}{1 + e^{x_i - \varepsilon}} = \frac{\prod_{i=1}^{n} e^{x_i - \varepsilon}}{\prod_{i=1}^{n} 1 + e^{x_i - \varepsilon}} = \frac{e^{\sum_{i=1}^{n} x_i - \varepsilon}}{\prod_{i=1}^{n} 1 + e^{x_i - \varepsilon}} = \frac{e^{-n\varepsilon}}{\prod_{i=1}^{n} 1 + e^{x_i - \varepsilon}}.$$

Similarly, we rewrite the right side as

$$\alpha^n = \sigma(-\varepsilon)^n = \left( \frac{e^{-\varepsilon}}{1 + e^{-\varepsilon}} \right)^n = \frac{e^{-n\varepsilon}}{(1 + e^{-\varepsilon})^n}.$$

Therefore, proving the left-side smaller or equal than the right-side, reduces to proving the left-denominator is larger than the right-denominator. Formally,

$$\prod_{i=1}^{n} 1 + e^{x_i - \varepsilon} \geq \left(1 + e^{-\varepsilon}\right)^n,$$

or equivalently, by applying the logarithmic function,

$$\sum_{i=1}^{n} \log(1 + e^{x_i - \varepsilon}) \geq n \log(1 + e^{-\varepsilon}). \tag{26}$$

To further ease the notation, we refer to the left side of inequality 26, as the target function

$$T(x) = \sum_{i=1}^{n} \log(1 + e^{x_i - \varepsilon}).$$

In particular, to prove 26, we show that

$$\min_{x} T(x) = n \log(1 + e^{-\varepsilon}), \tag{27}$$

for $x = \vec{0}$, which is the only stationary point due to the convexity of the target function.

Without loss of generality, we derive the partial derivative of the component $x_1$ on the target function $T(x)$. To constraint the components sum to zero, we consider the components $\{x_i\}$ for $i > 2$ as free, and then $x_2 = -x_1 - \sum_{i=3}^{n} x_i$ as a function of the remaining. The choice of $x_1, x_2$ is independent from the components ordering, and thus applies to any possible pair. Consequently,

$$\frac{\partial T(x)}{\partial x_1} = \frac{\partial(\log(1 + e^{x_1 - \varepsilon}) + \log(1 + e^{-x_1 - \sum_{i=3}^{n} x_i - \varepsilon}) + \sum_{i=3}^{n} \log(1 + e^{x_i - \varepsilon}))}{\partial x_1} \tag{28}$$

$$= \frac{\partial(\log(1 + e^{x_1 - \varepsilon}) + \log(1 + e^{-x_1 - \sum_{i=3}^{n} x_i - \varepsilon})}{\partial x_1} \tag{29}$$

$$= \sigma(x_1 - \varepsilon) - \sigma(-x_1 - \sum_{i=3}^{n} x_i - \varepsilon). \tag{30}$$

Since $\sigma(-\varepsilon) = \sigma(-\varepsilon)$, the equation is satisfied, for any component $x_i$, by $x = \vec{0}$,

We finally prove Inequality 26, by showing that the value of the target function $T(x)$, in its only stationary point $x = \vec{0}$, equals the bound. Formally,

$$T(\vec{0}) = \sum_{i=1}^{n} \log(1 + e^{-\varepsilon}) \tag{31}$$

$$= n \log(1 + e^{-\varepsilon}). \tag{32}$$

$\square$

**Lemma 3.** *(Sum of Differences in Cycle) Let $\{p_i\}$ be a sequence of $n + 1$ real numbers such that $p_1 = p_{n+1}$. Then, let $\{\delta_i\}$ be a sequence of $n$ numbers such that $\delta_i = p_{i+1} - p_i$. Then,*

$$\sum_{i=1}^{n} \delta_i = 0. \tag{33}$$

*Proof.* The proof is immediate from the following sequence of equations:

$$\sum_{i=1}^{n} \delta_i = \sum_{i=1}^{n} p_{i+1} - p_i = -p_1 + \sum_{i=2}^{n} p_i - p_i + p_{n+1} = 0.$$

$\square$

**Theorem 2.** Let $\mathbf{p} \in \mathbb{R}^d$ be a priority vector, $\varepsilon > 0$ be a strictly positive threshold, and $t > 0$ be a strictly positive temperature. Then, given the smooth orientation matrix $S_{t,\varepsilon}(\mathbf{p}) \in \mathbb{R}^{d \times d}$, it holds

$$h(S_{t,\varepsilon}(\mathbf{p})) \leq e^{d\alpha} - 1, \tag{34}$$

where $h(S_{t,\varepsilon}(\mathbf{p})) = \mathrm{tr}(e^{S_{t,\varepsilon}(\mathbf{p})}) - d$ is the NOTEARS acyclicity constraint and $\alpha = \sigma(-\varepsilon/t)$.

*Proof.* The left side of Inequality 34 corresponds to the following infinite series

$$\mathrm{tr}(e^{\mathbf{P}}) - d = \sum_{k=0}^{\infty} \frac{1}{k!} \mathrm{tr}(\mathbf{P}^{(k)}) - d$$

$$= \sum_{k=1}^{\infty} \frac{1}{k!} \mathrm{tr}(\mathbf{P}^{(k)})$$

where $\mathbf{P}^{(k)}$ is the matrix power defined as $\mathbf{P}^{(k)} = \mathbf{P}^{(k-1)}\mathbf{P}$ and $\mathbf{P}^0 = \mathbf{I}$.

By definition of matrix power, the $u$-th element on the diagonal of $\mathbf{P}^{(k)}$ equals to

$$\mathbf{P}_{uu}^{(k)} = \sum_{v_1 \in V} \mathbf{P}_{v_1,u}^{(k-1)} \mathbf{P}_{u,v_1}$$

$$= \sum_{v_1 \in V} \cdots \sum_{v_{k-1} \in V} \mathbf{P}_{u,v_1} \left( \prod_{i=1}^{k-2} \mathbf{P}_{v_i,v_{i+1}} \right) \mathbf{P}_{v_{k-1},u}.$$

Intuitively, the $u$-th element on the diagonal of $\mathbf{P}^{(k)}$ amounts to the sum of all possible paths starting and ending in the variable $X_u$. Therefore, being the same node, the priority of the first and the last node in the path are equal by definition. Consequently, by Lemma 3, the difference between the priorities sums to zero. For this reason, given Lemma 2, it holds that the product of the corresponding sigmoids is smaller or equal than $\alpha^k$. Therefore,

$$\mathbf{P}_{uu}^{(k)} = \sum_{v_1 \in V} \cdots \sum_{v_{k-1} \in V} \mathbf{P}_{u,v_1} \left( \prod_{i=1}^{k-2} \mathbf{P}_{v_i,v_{i+1}} \right) \mathbf{P}_{v_{k-1},u}$$

$$\leq \sum_{v_1 \in V} \cdots \sum_{v_{k-1} \in V} \alpha^k$$

$$= d^{k-1}\alpha^k.$$

Consequently, we upper bound the trace of the orientation matrix power as

$$\mathrm{tr}(\mathbf{P}^{(k)}) = \sum_{u=1}^{d} \mathbf{P}_{uu}^{(k)} \leq d^k \alpha^k.$$

Finally, we are able to prove the Theorem as

$$\mathrm{tr}(e^{\mathbf{P}}) - d = \sum_{k=0}^{\infty} \frac{1}{k!} \mathrm{tr}(\mathbf{P}^{(k)}) - d$$

$$= \sum_{k=1}^{\infty} \frac{1}{k!} \mathrm{tr}(\mathbf{P}^{(k)})$$

$$\leq \sum_{k=1}^{\infty} \frac{1}{k!} d^k \alpha^k$$

$$= -1 + e^{d\alpha},$$

where the last passage is due to the Taylor series of the exponential function. □

## C  Implementation Details

In this section, we discuss the significant aspects of our implementation. We run all the experiments on our internal cluster of Intel(R) Xeon(R) Gold 5120 processors, totaling 56 CPUs per machine. The necessary code and instructions are in the Supplementary Materials. We report details on the evaluation (C.1), the data generation procedure (C.2), and the models (C.4).

### C.1  Evaluation Procedure

We ensure a fair comparison by selecting the best hyperparameters for each implemented method on each dataset. We describe the hyperparameter space for each algorithm in the following subsections. Firstly, we sample fifty random configurations from the hyperparameter space. Since the hyperparameter space of COSMO also includes temperature and shift values, we extract more hyperparameters (200 – 800). Due to the significant speedup of COSMO, hyperparameter searches take a comparable amount of time, with NOTEARS being significantly longer on small graphs as well. Then, we test each configuration on five randomly sampled DAGs. We select the best hyperparameters according to the average AUC value. Finally, we perform a validation step by running the best configuration on five new random graphs.

To report the duration of each method, we track the time difference from the start of the fitting procedure up to the evaluation process, excluded. For constrained methods, this includes all the necessary adjustments between different problems. For COSMO, it contains the annealing of the temperature between training epochs.

Following previous work, we recover the binary adjacency matrix $\mathbf{A}$ of the retrieved graph by thresholding the learned weights $\mathbf{W}$ with a small constant $\omega = 0.3$. Formally, $\mathbf{A} = |\mathbf{W}| > \omega$.

### C.2  Synthetic Data

We include in our code the exact data generation process from the original implementation of NOTEARS.[2] Therefore, the dataset generation procedure firstly produces a DAG with either the Erdős–Rényi (ER) or the scale-free Barabási-Albert (SF) models. Then, it samples 1000 independent observations. In the linear case, the generator simulates equations of the form

$$f_i(x) = \mathbf{W}_i^\top x + z_i, \tag{35}$$

where we sample each weight $\mathbf{W}_{ij}$ from the uniform distribution $\mathcal{U}(-2, -0.5) \cup (0.5, 2)$ and each noise term $z_i$ from either the Normal, Exponential ($\lambda = 1$), or Gumbel ($\mu = 0, \beta = 1$) distributions. In the non-linear case, we simulate an additive noise model with form

$$f_i(x) = g_i(x) + z_i, \tag{36}$$

where $g_i$ is a randomly initialized Multilayer Perceptron (MLP) with 100 hidden units and the noise term $z_i$ is sampled from the Normal Distribution $\mathcal{N}(0, 1)$.

### C.3  Non-Linear Relations

We generalize COSMO to represent non-linear relations between variables. To ease the comparison, we follow the non-linear design of NOTEARS-MLP (Zheng et al., 2020) and DAGMA-MLP (Bello et al., 2022). However, it is worth mentioning that this is only one of the possible approaches for non-linear relations and that COSMO parameterization could be easily extended to mask the input of a neural network instead of the weights, as done by Ng et al. (2022b) or Brouillard et al. (2020). Similarly to NOTEARS (Zheng et al., 2020), we model the outcome of each variable $X_u$ with a neural network $f_u \colon \mathbb{R}^d \to \mathbb{R}$, where we distinguish between the first-layer weights $\mathbf{H}^u \in \mathbb{R}^{d \times h}$, for $h$ distinct neurons, and the remaining parameters $\Phi_u$. We ensure acyclicity by considering each weight matrix $\mathbf{H}^u$ as the $u$-th slice on the first dimension of a tensor $\mathbf{H} \in \mathbb{R}^{d \times d \times h}$. Intuitively, each entry $\mathbf{H}_{vi}^u$ represents the weight from the variable $X_u$ to the $i$-th neuron in the first layer of the

---

[2]NOTEARS implementation is published with Apache license at `https://github.com/xunzheng/notears`.

Table 4: Hyperparameter ranges and values for NOTEARS, NOCURL, and DAGMA.

| Hyperparameter | Range/Value |
|---|---|
| Learning Rate | (1e-3, 1e-2) |
| $\lambda_1$ | (1e-4, 1e-3) |
| $\lambda_2$ | (1e-3, 5e-3) |
| $\lambda_p$ | (1e-3, 3e-3) |
| $t_{\text{start}}$ | 0.45 |
| $t_{\text{end}}$ | (5e-4, 1e-3) |
| $\varepsilon$ | (5e-3, 2e-2) |

(a) COSMO

| Hyperparameter | Range/Value |
|---|---|
| Learning Rate | (1e-4, 1e-3) |
| Regularization | (1e-4, 1e-1) |
| Initial Penalty | 1.0 |
| Penalty Factor | 10.0 |
| Max Penalty | 1e+16 |
| Multiplier | 1.0 |
| DAGness Tolerance | 1e-8 |
| Progress Rate | 0.25 |
| Max Iterations | 100 |

(b) NOTEARS

| Hyperparameter | Range/Value |
|---|---|
| Learning Rate | (1e-4, 1e-3) |
| Regularization | (1e-4, 1e-1) |
| Initial Path Coefficient | 1.0 |
| Path Decay Factor | 0.1 |
| Log-Det Parameter | (1.0, 0.9, 0.8, 0.7) |
| Max Steps | 4 |

(c) DAGMA

| Hyperparameter | Range/Value |
|---|---|
| Learning Rate | (1e-4, 1e-3) |
| Regularization | (1e-4, 1e-1) |
| Multiplier | (10.0, 1000.0) |
| Prethreshold | 0.3 |
| Progress Rate | 0.25 |

(d) NOCURL

MLP $f_v$. Then, we broadcast the element-wise multiplication of a smooth orientation matrix on the hidden dimensions. Formally, we model the structural equation of each variable $X_u$ as

$$f_u(x) = g_u(\varphi(x^\top [\mathbf{H} \circ S_{t,\varepsilon}(\mathbf{p})]^u); \Phi_u), \tag{37}$$

where $\varphi$ is an activation function and $g_u$ is a Multilayer Perceptron (MLP) with weights $\Phi_u$. By applying each MLP $f_u$ to each variable $X_u$, we define the overall SEM as the function $f\colon \mathbb{R}^d \to \mathbb{R}^d$, which depends on the parameters $\mathbf{\Phi} = \{\Phi_u\}$, on the weight tensor $\mathbf{H}$, and the priority vector $\mathbf{p}$. Therefore, we formalize the non-linear extension of COSMO as the following problem

$$\min_{\mathbf{H} \in \mathbb{R}^{d \times d}, \mathbf{p} \in \mathbb{R}^d, \mathbf{\Phi}} \mathcal{L}(\mathbf{H} \circ S_{t,\varepsilon}(\mathbf{p}), \mathbf{\Phi}) + \lambda_1 \|\mathbf{H}\|_1 + \lambda_2 \|\mathbf{H}\|_2 + \lambda_p \|\mathbf{p}\|_2. \tag{38}$$

## C.4 MODELS

Since we focus on the role of acyclic learners as a building block within more comprehensive discovery solutions, we slightly detach from experimental setups considering such algorithms as standalone structure learners. Therefore, instead of dealing with full-batch optimization, we perform mini-batch optimization with batch size $B = 64$. Similarly, instead of explicitly computing the gradient of the loss function, we implement all methods in PyTorch to exploit automatic differentiation. By avoiding differentiation and other overhead sources, the time expenses results are not directly comparable between our implementations and the results reported in the original papers. However, our implementation choices are common to works that employed NOTEARS *et similia* to ensure the acyclicity of the solution (Lachapelle et al., 2020; Brouillard et al., 2020; Lopez et al., 2022).

By checking the convergence of the model, both NOTEARS, DAGMA, and NOCURL can dynamically stop the optimization procedure. On the other hand, COSMO requires a fixed number of epochs in which to anneal the temperature value. For a fair comparison, while we stop optimization problems after a maximum of 5000 training iterations, we do not disable early-stopping conditions on the baselines. Therefore, when sufficiently large, the maximum number of epochs should not affect the overall execution time of the methods. For COSMO, we interrupt the optimization after 2000 epochs. For the non-linear version of DAGMA, we increased the maximum epochs to 7000. Overall, we interrupt the execution of an algorithm whenever it hits a wall time limit of 20000 seconds.

As previously discussed in Subsection C.1, we perform a hyperparameter search on each model for each dataset. In particular, we sampled the learning rate from the range (1e-4, 1e-2) and the

regularization coefficients from the interval (1e-4, 1e-1). For the specific constrained optimization parameters, such as the number of problems or decay factors, we replicated the baseline parameters, for which we point the reader to the original papers or our implementation. For COSMO, we sample hyperparameters from the ranges in Table 4, given our theoretical findings on the relation between acyclicity and temperature (Theorem 2), we ensure sufficiently small acyclicity values. In the non-linear variant, we employ Multilayer Perceptrons with $h = 10$ hidden units for each variable.

## D DETAILED COMPARISON WITH RELATED WORKS

### D.1 COMPARISON WITH ENCO

Lippe et al. (2022) propose to learn a directed acyclic graph by jointly learning the probability of an arc being present and of the arc direction. The overall method, named ENCO, defines the probability of a direct edge $X_u \rightarrow X_v$ as

$$\mathbf{W}_{uv} = \sigma(\mathbf{H}_{uv}) \cdot \sigma(\mathbf{P}_{uv}), \tag{39}$$

where $\mathbf{H} \in \mathbb{R}^{d \times d}$ and $\mathbf{P} \in \mathbb{R}^{d \times d}$ are free parameters and $\sigma$ is the sigmoid function. Although similar to our formulation, ENCO defines arc orientations as a matrix that might be cyclic. In fact, the matrix $\mathbf{P}$ does not ensure the transitivity property that an orientation matrix grants instead. However, the authors proved that, in the limit of the number of samples from the interventional distribution, ENCO will converge to a directed acyclic graph. In comparison, our formulation always converges to a directed acyclic graph and is thus adapt to perform structure learning in the observational context.

### D.2 COMPARISON WITH NOCURL

Motivated by Hodge theory (Hodge, 1989), NOCURL (Yu et al., 2021) proposes to parameterize a DAG as a function of a $d$-dimensional vector and a directed possibly cyclic graph. By adopting our notation, introduced in Section 4, we could report their decomposition as

$$\mathbf{W}_{uv} = \mathbf{H}_{uv} \cdot \mathrm{ReLU}(\mathbf{p}_v - \mathbf{p}_u), \tag{40}$$

for an arbitrary weight from node $X_u$ to node $X_v$. Compared to our definition, the use ReLU does not correspond to the approximation of an orientation matrix. In fact, the distance between the priorities directly affects the weight. Instead, by employing the shifted-tempered sigmoid in the definition of smooth acyclic orientation, in COSMO the priorities only determine whether an arc is present between two variables. Further, NOCURL requires a preliminary solution from which to extract the topological ordering of the variables. In turn, such preliminary solution requires the use of an acyclicity constraint in multiple optimization problems. Therefore, in practice, NOCURL does not learn the variables ordering in an unconstrained way and adjusts adjacency weights in the last optimization problem.

## E ADDITIONAL RESULTS

In this section, we report further results on simulated DAGs with different noise terms, graph types, and increasing numbers of nodes. For each algorithm, we present the mean and standard deviation of each metric on five independent runs. We report the Area under the ROC Curve (AUC), the True Positive Ratio (TPR), and the Structural Hamming Distance normalized by the number of nodes (NHD). The reported duration of NOCURL includes the time to retrieve the necessary preliminary solution through two optimization problems regularized with the NOTEARS acyclicity constraint. We denote as NOCURL-U the variation of NOCURL that solves a unique unconstrained optimization problem without preliminary solution. When not immediate, we highlight in bold the **best** result and in italic bold the ***second best*** result. We do not report methods exceeding our wall time limit of 20000 seconds. For completeness, we report and discuss results on variance normalized datasets (Section E.18) and further comparisons with GOLEM (Section E.17) and DAGUERREOTYPE (Section E.19).

### E.1 ER4 - GAUSSIAN NOISE

| $d$ | Algorithm | NHD | TPR | AUC | Time (s) |
|---|---|---|---|---|---|
| 30 | COSMO | *0.867* ± *1.01* | **0.953** ± **0.04** | 0.984 ± 0.02 | **88** ± **3** |
| | DAGMA | **0.707** ± **0.57** | 0.940 ± 0.04 | **0.985** ± **0.01** | 781 ± 193 |
| | NOCURL | 1.653 ± 0.17 | *0.942* ± *0.02* | 0.967 ± 0.01 | 822 ± 15 |
| | NOCURL-U | 5.623 ± 0.92 | 0.492 ± 0.08 | 0.694 ± 0.06 | *227* ± *5* |
| | NOTEARS | 0.913 ± 0.60 | 0.940 ± 0.05 | 0.973 ± 0.02 | 5193 ± 170 |
| 100 | COSMO | *1.388* ± *0.69* | *0.917* ± *0.04* | 0.961 ± 0.03 | **99** ± **2** |
| | DAGMA | **1.026** ± **0.40** | 0.876 ± 0.02 | **0.982** ± **0.01** | 661 ± 142 |
| | NOCURL | 5.226 ± 1.34 | **0.921** ± **0.02** | 0.962 ± 0.01 | 1664 ± 15 |
| | NOCURL-U | 10.108 ± 4.11 | 0.427 ± 0.05 | 0.682 ± 0.05 | *267* ± *10* |
| | NOTEARS | 2.380 ± 2.10 | 0.898 ± 0.03 | *0.963* ± *0.01* | 11001 ± 340 |
| 500 | COSMO | *4.149* ± *1.14* | 0.819 ± 0.02 | *0.933* ± *0.01* | **437** ± **81** |
| | DAGMA | **2.246** ± **0.40** | **0.882** ± **0.01** | **0.980** ± **0.00** | 2485 ± 366 |
| | NOCURL-U | 27.675 ± 16.52 | 0.410 ± 0.04 | 0.683 ± 0.05 | *1546* ± *304* |

### E.2 ER4 - EXPONENTIAL NOISE

| $d$ | Algorithm | NHD | TPR | AUC | Time (s) |
|---|---|---|---|---|---|
| 30 | COSMO | **0.600** ± **0.54** | **0.970** ± **0.02** | **0.989** ± **0.01** | **89** ± **3** |
| | DAGMA | *0.613* ± *0.91* | *0.958* ± *0.05* | *0.986* ± *0.02* | 744 ± 75 |
| | NOCURL | 2.300 ± 0.97 | 0.918 ± 0.04 | 0.956 ± 0.02 | 826 ± 24 |
| | NOCURL-U | 5.313 ± 0.17 | 0.423 ± 0.05 | 0.694 ± 0.05 | *212* ± *5* |
| | NOTEARS | 1.320 ± 0.67 | 0.880 ± 0.10 | 0.966 ± 0.03 | 5579 ± 284 |
| 100 | COSMO | 1.642 ± 0.26 | **0.952** ± **0.02** | *0.985* ± *0.01* | **99** ± **2** |
| | DAGMA | *1.294* ± *0.52* | *0.944* ± *0.02* | **0.986** ± **0.01** | 733 ± 109 |
| | NOCURL | 5.652 ± 1.35 | 0.854 ± 0.03 | 0.950 ± 0.02 | 1655 ± 28 |
| | NOCURL-U | 11.642 ± 4.34 | 0.478 ± 0.05 | 0.693 ± 0.05 | *242* ± *4* |
| | NOTEARS | **1.156** ± **0.44** | 0.904 ± 0.03 | 0.972 ± 0.01 | 10880 ± 366 |
| 500 | COSMO | *2.342* ± *0.86* | **0.944** ± **0.02** | **0.986** ± **0.00** | **390** ± **102** |
| | DAGMA | **2.147** ± **1.08** | *0.902* ± *0.04* | *0.984* ± *0.01* | 2575 ± 469 |
| | NOCURL-U | 20.183 ± 7.43 | 0.437 ± 0.03 | 0.715 ± 0.03 | *1488* ± *249* |

### E.3 ER4 - GUMBEL NOISE

| $d$ | Algorithm | NHD | TPR | AUC | Time (s) |
|---|---|---|---|---|---|
| 30 | COSMO | 2.220 ± 1.65 | 0.862 ± 0.14 | 0.914 ± 0.10 | **87** ± **2** |
| | DAGMA | *1.680* ± *0.73* | *0.937* ± *0.03* | *0.973* ± *0.02* | 787 ± 86 |
| | NOCURL | 3.873 ± 1.26 | 0.853 ± 0.08 | 0.915 ± 0.04 | 826 ± 17 |
| | NOCURL-U | 5.260 ± 0.57 | 0.475 ± 0.08 | 0.678 ± 0.05 | *212* ± *5* |
| | NOTEARS | **0.587** ± **0.38** | **0.962** ± **0.03** | **0.981** ± **0.01** | 5229 ± 338 |
| 100 | COSMO | 2.398 ± 0.70 | **0.936** ± **0.02** | *0.973* ± *0.01* | **98** ± **1** |
| | DAGMA | **1.132** ± **0.79** | *0.921* ± *0.04* | **0.986** ± **0.01** | 858 ± 101 |
| | NOCURL | 4.714 ± 1.77 | 0.905 ± 0.03 | 0.962 ± 0.01 | 1675 ± 34 |
| | NOCURL-U | 6.914 ± 0.80 | 0.383 ± 0.04 | 0.663 ± 0.04 | *247* ± *9* |
| | NOTEARS | *1.402* ± *0.40* | 0.869 ± 0.04 | 0.969 ± 0.00 | 11889 ± 343 |
| 500 | COSMO | *3.574* ± *1.44* | **0.932** ± **0.02** | *0.982* ± *0.01* | **410** ± **106** |
| | DAGMA | **1.737** ± **0.64** | *0.871* ± *0.03* | **0.980** ± **0.00** | 2853 ± 218 |
| | NOCURL-U | 18.182 ± 9.28 | 0.462 ± 0.06 | 0.728 ± 0.05 | *1342* ± *209* |

### E.4 SF4 - GAUSSIAN NOISE

| $d$ | Algorithm | NHD | TPR | AUC | Time (s) |
|---|---|---|---|---|---|
| 30 | COSMO | **0.300 ± 0.09** | *0.973 ± 0.01* | **0.997 ± 0.00** | **89 ± 5** |
| | DAGMA | *0.360 ± 0.30* | **0.973 ± 0.02** | *0.996 ± 0.01* | 653 ± 198 |
| | NOCURL | 0.967 ± 0.43 | 0.893 ± 0.03 | 0.983 ± 0.01 | 828 ± 23 |
| | NOCURL-U | 4.410 ± 0.72 | 0.566 ± 0.11 | 0.741 ± 0.08 | *226 ± 7* |
| | NOTEARS | 0.553 ± 0.54 | 0.944 ± 0.06 | 0.984 ± 0.02 | 5292 ± 261 |
| 100 | COSMO | *0.482 ± 0.31* | *0.962 ± 0.02* | 0.991 ± 0.01 | **99 ± 3** |
| | DAGMA | 0.712 ± 0.33 | 0.951 ± 0.02 | **0.995 ± 0.00** | 479 ± 75 |
| | NOCURL | 2.030 ± 0.46 | 0.883 ± 0.03 | 0.982 ± 0.01 | 1667 ± 25 |
| | NOCURL-U | 5.521 ± 0.61 | 0.596 ± 0.09 | 0.788 ± 0.06 | *269 ± 9* |
| | NOTEARS | **0.280 ± 0.35** | **0.972 ± 0.04** | *0.993 ± 0.01* | 10112 ± 492 |
| 500 | COSMO | *1.566 ± 0.68* | **0.953 ± 0.02** | *0.989 ± 0.01* | **541 ± 15** |
| | DAGMA | **1.343 ± 0.46** | *0.915 ± 0.04* | **0.992 ± 0.00** | *1345 ± 33* |
| | NOCURL-U | 7.146 ± 3.19 | 0.504 ± 0.08 | 0.780 ± 0.07 | 1394 ± 217 |

### E.5 SF4 - EXPONENTIAL NOISE

| $d$ | Algorithm | NHD | TPR | AUC | Time (s) |
|---|---|---|---|---|---|
| 30 | COSMO | 0.613 ± 0.39 | *0.965 ± 0.02* | 0.985 ± 0.02 | **87 ± 2** |
| | DAGMA | **0.127 ± 0.20** | **0.991 ± 0.01** | **0.999 ± 0.00** | 592 ± 200 |
| | NOCURL | 0.887 ± 0.21 | 0.845 ± 0.02 | *0.985 ± 0.01* | 824 ± 25 |
| | NOCURL-U | 4.067 ± 0.73 | 0.460 ± 0.15 | 0.685 ± 0.09 | *212 ± 7* |
| | NOTEARS | *0.513 ± 0.30* | 0.962 ± 0.03 | 0.984 ± 0.01 | 5189 ± 271 |
| 100 | COSMO | *0.724 ± 0.71* | *0.963 ± 0.04* | 0.985 ± 0.02 | **100 ± 2** |
| | DAGMA | **0.586 ± 0.56** | **0.969 ± 0.03** | **0.995 ± 0.00** | 395 ± 108 |
| | NOCURL | 1.998 ± 0.40 | 0.907 ± 0.03 | 0.980 ± 0.00 | 1670 ± 28 |
| | NOCURL-U | 5.912 ± 1.54 | 0.575 ± 0.06 | 0.783 ± 0.04 | *245 ± 7* |
| | NOTEARS | 0.910 ± 0.43 | 0.962 ± 0.02 | *0.991 ± 0.01* | 10243 ± 723 |
| 500 | COSMO | **1.445 ± 0.58** | **0.950 ± 0.03** | **0.990 ± 0.01** | **517 ± 108** |
| | DAGMA | *1.653 ± 0.91* | *0.873 ± 0.08* | *0.988 ± 0.01* | 1466 ± 247 |
| | NOCURL-U | 12.140 ± 7.84 | 0.482 ± 0.08 | 0.727 ± 0.06 | *1205 ± 257* |

### E.6 SF4 - GUMBEL NOISE

| $d$ | Algorithm | NHD | TPR | AUC | Time (s) |
|---|---|---|---|---|---|
| 30 | COSMO | **0.467 ± 0.51** | **0.962 ± 0.05** | *0.990 ± 0.02* | **88 ± 2** |
| | DAGMA | *0.487 ± 0.20* | *0.956 ± 0.03* | **0.990 ± 0.01** | 754 ± 179 |
| | NOCURL | 0.747 ± 0.19 | 0.938 ± 0.02 | 0.989 ± 0.00 | 826 ± 32 |
| | NOCURL-U | 3.107 ± 0.64 | 0.460 ± 0.06 | 0.737 ± 0.04 | *213 ± 5* |
| | NOTEARS | 0.860 ± 0.76 | 0.924 ± 0.06 | 0.975 ± 0.02 | 5199 ± 130 |
| 100 | COSMO | *0.864 ± 0.24* | *0.968 ± 0.01* | *0.992 ± 0.01* | **98 ± 2** |
| | DAGMA | **0.388 ± 0.30** | **0.975 ± 0.02** | **0.997 ± 0.00** | 422 ± 103 |
| | NOCURL | 1.806 ± 0.40 | 0.898 ± 0.03 | 0.982 ± 0.01 | 1676 ± 31 |
| | NOCURL-U | 8.756 ± 2.65 | 0.550 ± 0.05 | 0.757 ± 0.03 | *245 ± 7* |
| | NOTEARS | 1.134 ± 0.81 | 0.894 ± 0.08 | 0.989 ± 0.01 | 11618 ± 1309 |
| 500 | COSMO | *1.426 ± 0.53* | **0.951 ± 0.03** | **0.994 ± 0.00** | **524 ± 22** |
| | DAGMA | **1.384 ± 0.38** | *0.849 ± 0.04* | *0.991 ± 0.00* | 1359 ± 34 |
| | NOCURL-U | 8.931 ± 7.05 | 0.430 ± 0.10 | 0.741 ± 0.08 | *1193 ± 229* |

### E.7  ER6 - GAUSSIAN NOISE

| $d$ | Algorithm | NHD | TPR | AUC | Time (s) |
|---|---|---|---|---|---|
| 30 | COSMO | $4.087 \pm 1.12$ | $0.838 \pm 0.06$ | $0.921 \pm 0.04$ | $\mathbf{89 \pm 4}$ |
| | DAGMA | $\mathbf{2.367 \pm 0.63}$ | $\mathit{0.847 \pm 0.03}$ | $\mathbf{0.958 \pm 0.01}$ | $665 \pm 249$ |
| | NOCURL | $4.480 \pm 0.92$ | $\mathbf{0.869 \pm 0.03}$ | $0.908 \pm 0.03$ | $909 \pm 18$ |
| | NOCURL-U | $7.490 \pm 1.18$ | $0.459 \pm 0.08$ | $0.672 \pm 0.06$ | $\mathit{226 \pm 6}$ |
| | NOTEARS | $\mathit{3.327 \pm 1.65}$ | $0.840 \pm 0.07$ | $\mathit{0.922 \pm 0.04}$ | $5239 \pm 427$ |
| 100 | COSMO | $\mathit{9.476 \pm 3.01}$ | $0.771 \pm 0.08$ | $\mathit{0.911 \pm 0.05}$ | $\mathbf{98 \pm 2}$ |
| | DAGMA | $10.740 \pm 2.83$ | $0.709 \pm 0.13$ | $0.902 \pm 0.04$ | $761 \pm 134$ |
| | NOCURL | $15.044 \pm 1.60$ | $\mathit{0.785 \pm 0.04}$ | $0.888 \pm 0.02$ | $1687 \pm 26$ |
| | NOCURL-U | $30.719 \pm 5.20$ | $0.435 \pm 0.03$ | $0.580 \pm 0.04$ | $\mathit{268 \pm 9}$ |
| | NOTEARS | $\mathbf{6.556 \pm 3.10}$ | $\mathbf{0.842 \pm 0.05}$ | $\mathbf{0.944 \pm 0.02}$ | $12053 \pm 940$ |
| 500 | COSMO | $\mathit{25.443 \pm 4.47}$ | $\mathbf{0.736 \pm 0.01}$ | $\mathbf{0.937 \pm 0.01}$ | $\mathbf{526 \pm 100}$ |
| | DAGMA | $\mathbf{15.952 \pm 1.67}$ | $\mathit{0.553 \pm 0.05}$ | $\mathit{0.925 \pm 0.01}$ | $3207 \pm 271$ |
| | NOCURL-U | $165.465 \pm 20.86$ | $0.433 \pm 0.02$ | $0.558 \pm 0.03$ | $\mathit{1226 \pm 293}$ |

### E.8  ER6 - EXPONENTIAL NOISE

| $d$ | Algorithm | NHD | TPR | AUC | Time (s) |
|---|---|---|---|---|---|
| 30 | COSMO | $\mathit{3.300 \pm 0.95}$ | $\mathbf{0.897 \pm 0.05}$ | $\mathit{0.947 \pm 0.03}$ | $\mathbf{89 \pm 2}$ |
| | DAGMA | $3.480 \pm 1.42$ | $0.861 \pm 0.06$ | $0.945 \pm 0.03$ | $672 \pm 177$ |
| | NOCURL | $4.573 \pm 0.78$ | $0.846 \pm 0.05$ | $0.902 \pm 0.03$ | $897 \pm 13$ |
| | NOCURL-U | $8.700 \pm 0.89$ | $0.426 \pm 0.07$ | $0.615 \pm 0.06$ | $\mathit{226 \pm 9}$ |
| | NOTEARS | $\mathbf{2.313 \pm 1.55}$ | $\mathit{0.881 \pm 0.09}$ | $\mathbf{0.953 \pm 0.04}$ | $5516 \pm 652$ |
| 100 | COSMO | $10.170 \pm 2.74$ | $0.768 \pm 0.09$ | $0.919 \pm 0.04$ | $\mathbf{99 \pm 3}$ |
| | DAGMA | $\mathit{8.118 \pm 3.10}$ | $\mathit{0.793 \pm 0.11}$ | $\mathit{0.934 \pm 0.04}$ | $681 \pm 149$ |
| | NOCURL | $14.860 \pm 4.67$ | $0.685 \pm 0.10$ | $0.863 \pm 0.06$ | $1735 \pm 39$ |
| | NOCURL-U | $30.600 \pm 4.34$ | $0.450 \pm 0.04$ | $0.591 \pm 0.04$ | $\mathit{267 \pm 8}$ |
| | NOTEARS | $\mathbf{5.208 \pm 2.54}$ | $\mathbf{0.796 \pm 0.09}$ | $\mathbf{0.943 \pm 0.03}$ | $12663 \pm 1555$ |
| 500 | COSMO | $\mathit{25.854 \pm 4.28}$ | $\mathbf{0.741 \pm 0.04}$ | $\mathbf{0.943 \pm 0.01}$ | $\mathbf{460 \pm 123}$ |
| | DAGMA | $\mathbf{16.417 \pm 4.45}$ | $\mathit{0.571 \pm 0.11}$ | $\mathit{0.925 \pm 0.02}$ | $4069 \pm 580$ |
| | NOCURL-U | $152.336 \pm 31.97$ | $0.425 \pm 0.02$ | $0.567 \pm 0.03$ | $\mathit{1363 \pm 306}$ |

### E.9  ER6 - GUMBEL NOISE

| $d$ | Algorithm | NHD | TPR | AUC | Time (s) |
|---|---|---|---|---|---|
| 30 | COSMO | $2.840 \pm 1.08$ | $\mathbf{0.906 \pm 0.04}$ | $\mathit{0.954 \pm 0.03}$ | $\mathbf{89 \pm 3}$ |
| | DAGMA | $\mathbf{2.727 \pm 0.83}$ | $\mathit{0.906 \pm 0.02}$ | $\mathbf{0.964 \pm 0.02}$ | $634 \pm 194$ |
| | NOCURL | $5.003 \pm 0.72$ | $0.811 \pm 0.04$ | $0.891 \pm 0.03$ | $902 \pm 9$ |
| | NOCURL-U | $8.153 \pm 0.96$ | $0.422 \pm 0.07$ | $0.629 \pm 0.04$ | $\mathit{226 \pm 6}$ |
| | NOTEARS | $\mathit{2.740 \pm 1.61}$ | $0.791 \pm 0.10$ | $0.938 \pm 0.04$ | $5416 \pm 446$ |
| 100 | COSMO | $10.048 \pm 3.15$ | $0.780 \pm 0.07$ | $0.899 \pm 0.06$ | $\mathbf{100 \pm 3}$ |
| | DAGMA | $\mathit{7.910 \pm 3.05}$ | $\mathit{0.805 \pm 0.09}$ | $\mathit{0.935 \pm 0.04}$ | $715 \pm 203$ |
| | NOCURL | $11.932 \pm 2.68$ | $0.742 \pm 0.04$ | $0.894 \pm 0.03$ | $1688 \pm 34$ |
| | NOCURL-U | $27.401 \pm 4.42$ | $0.431 \pm 0.05$ | $0.600 \pm 0.04$ | $\mathit{266 \pm 4}$ |
| | NOTEARS | $\mathbf{4.884 \pm 0.45}$ | $\mathbf{0.833 \pm 0.05}$ | $\mathbf{0.951 \pm 0.01}$ | $12634 \pm 639$ |
| 500 | COSMO | $\mathit{26.148 \pm 4.86}$ | $\mathbf{0.740 \pm 0.04}$ | $\mathbf{0.941 \pm 0.02}$ | $\mathbf{418 \pm 106}$ |
| | DAGMA | $\mathbf{16.358 \pm 4.94}$ | $\mathit{0.563 \pm 0.07}$ | $\mathit{0.921 \pm 0.02}$ | $3527 \pm 241$ |
| | NOCURL-U | $125.858 \pm 36.61$ | $0.367 \pm 0.06$ | $0.571 \pm 0.02$ | $\mathit{1612 \pm 27}$ |

### E.10  SF6 - GAUSSIAN NOISE

| $d$ | Algorithm | NHD | TPR | AUC | Time (s) |
|---|---|---|---|---|---|
| 30 | COSMO | $1.273 \pm 1.07$ | $0.907 \pm 0.10$ | $0.963 \pm 0.06$ | $\mathbf{89 \pm 2}$ |
| | DAGMA | $\mathit{1.107 \pm 0.37}$ | $\mathbf{0.930 \pm 0.03}$ | $\mathbf{0.985 \pm 0.01}$ | $456 \pm 39$ |
| | NOCURL | $1.573 \pm 0.46$ | $0.864 \pm 0.04$ | $0.973 \pm 0.01$ | $823 \pm 14$ |
| | NOCURL-U | $4.997 \pm 0.98$ | $0.506 \pm 0.05$ | $0.732 \pm 0.05$ | $\mathit{226 \pm 8}$ |
| | NOTEARS | $\mathbf{0.933 \pm 0.71}$ | $\mathit{0.919 \pm 0.05}$ | $\mathit{0.984 \pm 0.02}$ | $5313 \pm 184$ |
| 100 | COSMO | $4.478 \pm 2.88$ | $0.776 \pm 0.15$ | $0.874 \pm 0.11$ | $\mathbf{99 \pm 2}$ |
| | DAGMA | $\mathit{2.024 \pm 0.71}$ | $\mathit{0.914 \pm 0.02}$ | $\mathit{0.987 \pm 0.00}$ | $396 \pm 53$ |
| | NOCURL | $2.824 \pm 0.39$ | $0.818 \pm 0.02$ | $0.980 \pm 0.00$ | $1679 \pm 27$ |
| | NOCURL-U | $10.556 \pm 6.00$ | $0.542 \pm 0.07$ | $0.751 \pm 0.08$ | $\mathit{266 \pm 5}$ |
| | NOTEARS | $\mathbf{1.412 \pm 0.59}$ | $\mathbf{0.939 \pm 0.03}$ | $\mathbf{0.990 \pm 0.01}$ | $11156 \pm 170$ |
| 500 | COSMO | $\mathit{4.670 \pm 1.99}$ | $\mathbf{0.912 \pm 0.02}$ | $\mathbf{0.984 \pm 0.00}$ | $\mathbf{460 \pm 70}$ |
| | DAGMA | $\mathbf{3.825 \pm 0.19}$ | $\mathit{0.746 \pm 0.03}$ | $\mathit{0.982 \pm 0.00}$ | $1418 \pm 54$ |
| | NOCURL-U | $19.793 \pm 11.03$ | $0.368 \pm 0.04$ | $0.698 \pm 0.04$ | $\mathit{1137 \pm 231}$ |

### E.11  SF6 - EXPONENTIAL NOISE

| $d$ | Algorithm | NHD | TPR | AUC | Time (s) |
|---|---|---|---|---|---|
| 30 | COSMO | $1.393 \pm 1.24$ | $0.926 \pm 0.05$ | $0.975 \pm 0.03$ | $\mathbf{88 \pm 1}$ |
| | DAGMA | $\mathit{1.147 \pm 0.48}$ | $\mathit{0.943 \pm 0.03}$ | $\mathit{0.982 \pm 0.01}$ | $578 \pm 173$ |
| | NOCURL | $1.987 \pm 0.54$ | $0.757 \pm 0.08$ | $0.967 \pm 0.01$ | $820 \pm 8$ |
| | NOCURL-U | $4.787 \pm 0.99$ | $0.534 \pm 0.07$ | $0.761 \pm 0.06$ | $\mathit{227 \pm 7}$ |
| | NOTEARS | $\mathbf{0.753 \pm 0.49}$ | $\mathbf{0.943 \pm 0.04}$ | $\mathbf{0.986 \pm 0.01}$ | $5312 \pm 258$ |
| 100 | COSMO | $3.836 \pm 2.75$ | $0.864 \pm 0.09$ | $0.944 \pm 0.05$ | $\mathbf{98 \pm 2}$ |
| | DAGMA | $\mathbf{1.532 \pm 0.61}$ | $0.887 \pm 0.04$ | $\mathbf{0.988 \pm 0.00}$ | $373 \pm 88$ |
| | NOCURL | $2.890 \pm 0.61$ | $\mathit{0.910 \pm 0.02}$ | $0.977 \pm 0.00$ | $1692 \pm 28$ |
| | NOCURL-U | $6.607 \pm 1.05$ | $0.474 \pm 0.06$ | $0.760 \pm 0.06$ | $\mathit{266 \pm 2}$ |
| | NOTEARS | $\mathit{1.784 \pm 0.52}$ | $\mathbf{0.939 \pm 0.02}$ | $\mathit{0.988 \pm 0.00}$ | $11369 \pm 519$ |
| 500 | COSMO | $\mathbf{3.144 \pm 0.47}$ | $\mathbf{0.919 \pm 0.02}$ | $\mathbf{0.989 \pm 0.00}$ | $\mathbf{457 \pm 81}$ |
| | DAGMA | $\mathit{3.854 \pm 0.34}$ | $\mathit{0.750 \pm 0.01}$ | $\mathit{0.977 \pm 0.01}$ | $\mathit{1384 \pm 33}$ |
| | NOCURL-U | $13.763 \pm 8.79$ | $0.389 \pm 0.05$ | $0.728 \pm 0.06$ | $1436 \pm 230$ |

### E.12  SF6 - GUMBEL NOISE

| $d$ | Algorithm | NHD | TPR | AUC | Time (s) |
|---|---|---|---|---|---|
| 30 | COSMO | $\mathbf{1.047 \pm 0.42}$ | $\mathbf{0.938 \pm 0.03}$ | $\mathbf{0.984 \pm 0.01}$ | $\mathbf{88 \pm 1}$ |
| | DAGMA | $1.347 \pm 0.63$ | $\mathit{0.933 \pm 0.02}$ | $\mathit{0.981 \pm 0.01}$ | $528 \pm 67$ |
| | NOCURL | $1.787 \pm 0.52$ | $0.898 \pm 0.02$ | $0.969 \pm 0.01$ | $822 \pm 29$ |
| | NOCURL-U | $5.577 \pm 0.43$ | $0.549 \pm 0.06$ | $0.733 \pm 0.05$ | $\mathit{225 \pm 4}$ |
| | NOTEARS | $\mathit{1.053 \pm 0.59}$ | $0.911 \pm 0.04$ | $0.978 \pm 0.02$ | $5429 \pm 251$ |
| 100 | COSMO | $3.486 \pm 2.62$ | $0.879 \pm 0.10$ | $0.947 \pm 0.06$ | $\mathbf{99 \pm 2}$ |
| | DAGMA | $\mathbf{1.418 \pm 0.34}$ | $\mathit{0.910 \pm 0.03}$ | $\mathbf{0.990 \pm 0.00}$ | $424 \pm 90$ |
| | NOCURL | $3.074 \pm 0.50$ | $0.893 \pm 0.02$ | $0.976 \pm 0.00$ | $1682 \pm 22$ |
| | NOCURL-U | $9.643 \pm 4.59$ | $0.464 \pm 0.08$ | $0.712 \pm 0.10$ | $\mathit{267 \pm 9}$ |
| | NOTEARS | $\mathit{1.586 \pm 1.39}$ | $0.913 \pm 0.06$ | $\mathit{0.987 \pm 0.01}$ | $11820 \pm 985$ |
| 500 | COSMO | $\mathbf{3.288 \pm 0.50}$ | $\mathbf{0.931 \pm 0.01}$ | $\mathbf{0.992 \pm 0.00}$ | $429 \pm 87$ |
| | DAGMA | $\mathit{4.055 \pm 0.88}$ | $\mathit{0.802 \pm 0.03}$ | $\mathit{0.981 \pm 0.00}$ | $1465 \pm 138$ |
| | NOCURL-U | $56.103 \pm 41.06$ | $0.420 \pm 0.06$ | $0.648 \pm 0.07$ | $\mathit{1201 \pm 253}$ |

### E.13 ER4 - NON-LINEAR MLP

| $d$ | Algorithm | NHD | TPR | AUC | Time (s) |
|---|---|---|---|---|---|
| 20 | COSMO | $2.530 \pm 0.45$ | $0.752 \pm 0.06$ | $0.923 \pm 0.03$ | $154 \pm 2$ |
| | DAGMA-MLP | $2.853 \pm 0.37$ | $0.810 \pm 0.15$ | $0.932 \pm 0.04$ | $2252 \pm 40$ |
| | NOTEARS-MLP | $3.270 \pm 0.52$ | $0.904 \pm 0.07$ | $0.956 \pm 0.02$ | $3554 \pm 154$ |
| 40 | COSMO | $3.295 \pm 0.45$ | $0.712 \pm 0.05$ | $0.925 \pm 0.02$ | $177 \pm 2$ |
| | DAGMA-MLP | $3.633 \pm 0.81$ | $0.804 \pm 0.10$ | $0.942 \pm 0.03$ | $2622 \pm 28$ |
| 100 | COSMO | $5.164 \pm 1.38$ | $0.582 \pm 0.04$ | $0.895 \pm 0.02$ | $311 \pm 10$ |
| | DAGMA-MLP | $2.337 \pm 0.18$ | $0.506 \pm 0.05$ | $0.912 \pm 0.01$ | $4866 \pm 69$ |

### E.14 SF4 - NON-LINEAR MLP

| $d$ | Algorithm | NHD | TPR | AUC | Time (s) |
|---|---|---|---|---|---|
| 20 | COSMO | $1.525 \pm 0.18$ | $0.743 \pm 0.05$ | $0.954 \pm 0.03$ | $155 \pm 3$ |
| | DAGMA-MLP | $1.408 \pm 0.26$ | $0.746 \pm 0.11$ | $0.968 \pm 0.01$ | $2270 \pm 27$ |
| | NOTEARS-MLP | $1.040 \pm 0.33$ | $0.922 \pm 0.08$ | $0.980 \pm 0.02$ | $3400 \pm 184$ |
| 40 | COSMO | $2.271 \pm 0.24$ | $0.583 \pm 0.05$ | $0.953 \pm 0.01$ | $174 \pm 5$ |
| | DAGMA-MLP | $1.875 \pm 0.34$ | $0.744 \pm 0.10$ | $0.972 \pm 0.01$ | $2588 \pm 45$ |
| 100 | COSMO | $3.144 \pm 0.23$ | $0.455 \pm 0.04$ | $0.945 \pm 0.01$ | $313 \pm 6$ |
| | DAGMA-MLP | $3.051 \pm 0.14$ | $0.260 \pm 0.04$ | $0.961 \pm 0.01$ | $4883 \pm 77$ |

### E.15 ER6 - NON-LINEAR MLP

| $d$ | Algorithm | NHD | TPR | AUC | Time (s) |
|---|---|---|---|---|---|
| 20 | COSMO | $2.995 \pm 0.45$ | $0.667 \pm 0.07$ | $0.919 \pm 0.03$ | $155 \pm 2$ |
| | DAGMA-MLP | $2.992 \pm 0.39$ | $0.712 \pm 0.08$ | $0.917 \pm 0.02$ | $2252 \pm 35$ |
| | NOTEARS-MLP | $2.895 \pm 0.47$ | $0.862 \pm 0.08$ | $0.949 \pm 0.02$ | $3557 \pm 130$ |
| 40 | COSMO | $4.837 \pm 0.66$ | $0.598 \pm 0.07$ | $0.891 \pm 0.02$ | $174 \pm 3$ |
| | DAGMA-MLP | $3.732 \pm 0.71$ | $0.633 \pm 0.13$ | $0.919 \pm 0.02$ | $2570 \pm 44$ |
| 100 | COSMO | $6.049 \pm 0.91$ | $0.501 \pm 0.04$ | $0.875 \pm 0.02$ | $308 \pm 4$ |
| | DAGMA-MLP | $3.790 \pm 0.24$ | $0.535 \pm 0.05$ | $0.908 \pm 0.01$ | $4807 \pm 80$ |

### E.16 SF6 - NON-LINEAR MLP

| $d$ | Algorithm | NHD | TPR | AUC | Time (s) |
|---|---|---|---|---|---|
| 20 | COSMO | $1.855 \pm 0.28$ | $0.748 \pm 0.06$ | $0.959 \pm 0.01$ | $154 \pm 2$ |
| | DAGMA-MLP | $1.833 \pm 0.52$ | $0.766 \pm 0.13$ | $0.957 \pm 0.03$ | $2249 \pm 40$ |
| | NOTEARS-MLP | $1.355 \pm 0.39$ | $0.926 \pm 0.08$ | $0.980 \pm 0.01$ | $3512 \pm 193$ |
| 40 | COSMO | $3.394 \pm 0.25$ | $0.501 \pm 0.06$ | $0.946 \pm 0.01$ | $174 \pm 4$ |
| | DAGMA-MLP | $2.720 \pm 0.38$ | $0.724 \pm 0.08$ | $0.967 \pm 0.01$ | $2572 \pm 100$ |
| 100 | COSMO | $4.744 \pm 0.21$ | $0.342 \pm 0.04$ | $0.931 \pm 0.01$ | $308 \pm 6$ |
| | DAGMA-MLP | $4.149 \pm 0.17$ | $0.391 \pm 0.06$ | $0.968 \pm 0.01$ | $4699 \pm 338$ |

| Graph | $d$ | Algorithm | NHD | TPR | AUC | $h(W)$ | Time (s) |
|---|---|---|---|---|---|---|---|
| ER4 | 30 | COSMO | $0.867 \pm 1.01$ | $\mathbf{0.953 \pm 0.04}$ | $\mathbf{0.984 \pm 0.02}$ | $\mathbf{0.000 \pm 0.00}$ | $\mathbf{88 \pm 3}$ |
| | | GOLEM | $\mathbf{0.833 \pm 1.14}$ | $0.933 \pm 0.09$ | $0.975 \pm 0.04$ | $0.025 \pm 0.03$ | $305 \pm 4$ |
| | 100 | COSMO | $\mathbf{1.388 \pm 0.69}$ | $0.917 \pm 0.04$ | $0.961 \pm 0.03$ | $\mathbf{0.000 \pm 0.00}$ | $\mathbf{99 \pm 2}$ |
| | | GOLEM | $4.874 \pm 1.02$ | $0.802 \pm 0.04$ | $0.936 \pm 0.01$ | $0.548 \pm 0.22$ | $660 \pm 11$ |
| ER6 | 30 | COSMO | $\mathbf{4.087 \pm 1.12}$ | $\mathbf{0.838 \pm 0.06}$ | $\mathbf{0.921 \pm 0.04}$ | $\mathbf{0.000 \pm 0.00}$ | $\mathbf{89 \pm 4}$ |
| | | GOLEM | $6.153 \pm 0.56$ | $0.496 \pm 0.13$ | $0.749 \pm 0.10$ | $0.382 \pm 0.14$ | $306 \pm 9$ |
| | 100 | COSMO | $\mathbf{9.476 \pm 3.01}$ | $0.771 \pm 0.08$ | $0.911 \pm 0.05$ | $\mathbf{0.000 \pm 0.00}$ | $\mathbf{98 \pm 2}$ |
| | | GOLEM | $9.572 \pm 2.32$ | $0.423 \pm 0.08$ | $0.794 \pm 0.07$ | $4.654 \pm 1.34$ | $664 \pm 9$ |

Table 5: Experimental comparison with GOLEM (Ng et al., a).

### E.17 GOLEM (NG ET AL., A)

GOLEM (Ng et al., a) is an unconstrained approach for continuous structure learning that employs the NOTEARS acylicity constraint as a regularization term. Given this strategy, GOLEM does not ensure DAG convergence and instead requires a final pruning step to iteratively remove arcs until the resulting adjacency matrix is acyclic. As stated in the Related Works (Section 2), we focus our main empirical comparison on methods ensuring the acyclicity of the solution. Therefore, in the following table, we only briefly report an empirical analysis of GOLEM. Notably, on small and sparse graphs ($d = 30$, ER4), the resulting acyclicity might be tolerable, but the value is significant on both larger ($d = 100$) and denser (ER6) ones. We also report COSMO results on the same graph types to recall that our proposal achieves acyclicity zero by construction (Table 5).

### E.18 NORMALIZED VARIANCE RESULTS

As we remarked in the main body, the choice of optimizing an adjacency matrix by minimizing the MSE is known to exploit the variance of the variables (Loh & Bühlmann, 2014) and might not be significant outside of this assumption (Reisach et al., 2021; Kaiser & Sipos, 2022; Ng et al., b). In general, variance normalization is enough to induce a significant performance drop in approaches adopting this loss function. However, it is worth remarking that structure learning formulations derived from NOTEARS, like ours, are loss-agnostic and have been used in several continuous causal discovery methods with different but appropriate loss functions (Brouillard et al., 2020; Lorch et al., 2022; Hägele et al., 2023). Defining function classes, appropriate losses, and identifiability results are important requirements for causal discovery solutions, but are out of the scope of the current work. However, for clarity, we show that, when fitting a DAG through the MSE loss, COSMO is equally affected by variance normalization as competing approaches (Table 6).

Reisach et al. (2021) proposes a simple baseline named SORTNREGRESS to evaluate whether the variables order follow their variance in non-normalized datasets. The algorithm consists of two steps: (i.) sorting the nodes according to their variance and (ii.) regressing variables on their predecessors. In datasets where there is a strict relation between the topological ordering and the nodes' variance, the SORTNREGRESS baseline is expected to perform consistently well. To better contextualize our work, we show that in the simulated test bed adopted by NOTEARS, DAGMA, NOCURL, and COSMO, the baseline performs well as expected (Table 7). Notably, it is evident how, for larger graphs, our parameterization is still faster the SORTNREGRESS baseline, despite the simplicity of the latter.

### E.19 DAGUERREOTYPE (ZANTEDESCHI ET AL., 2022)

DAGUERREOTYPE (Zantedeschi et al., 2022) proposes an end-to-end differentiable model for causal DAG discovery that parameterize the underlying causal ordering over the polytope of permutation vectors. In practice, the model fits a distribution over possible permutations parameterized by a score vector analogous to the priority vector $\mathbf{p}$ used by COSMO. The differentiable transformation from the score vector to the permutation can be realized by either the SparseMAP (Niculae et al., 2018) or the Top-$k$ sparsemax (Correia et al., 2020) operators. Further, the scores and the functional relations — adjacencies in the linear case — can be either learned jointly or with a bi-level optimization scheme.

| Graph | $d$ | Algorithm | NHD | TPR | AUC | $h(W)$ |
|---|---|---|---|---|---|---|
| ER4 | 30 | COSMO | **4.637** $\pm$ **0.43** | 0.095 $\pm$ 0.04 | 0.440 $\pm$ 0.06 | **0.000** $\pm$ **0.00** |
| | | DAGMA | 4.967 $\pm$ 0.41 | **0.102** $\pm$ **0.04** | *0.475* $\pm$ *0.08* | *0.000* $\pm$ *0.00* |
| | | NOCURL | *4.910* $\pm$ *0.53* | 0.059 $\pm$ 0.04 | 0.429 $\pm$ 0.07 | 0.000 $\pm$ 0.00 |
| | | NOCURL-U | 4.950 $\pm$ 0.31 | 0.092 $\pm$ 0.03 | **0.530** $\pm$ **0.05** | 0.000 $\pm$ 0.00 |
| | | NOTEARS | 4.920 $\pm$ 0.36 | *0.098* $\pm$ *0.04* | 0.473 $\pm$ 0.06 | 0.000 $\pm$ 0.00 |
| ER4 | 100 | COSMO | **4.150** $\pm$ **0.05** | 0.036 $\pm$ 0.01 | 0.533 $\pm$ 0.02 | **0.000** $\pm$ **0.00** |
| | | DAGMA | *4.205* $\pm$ *0.10* | *0.120* $\pm$ *0.02* | *0.571* $\pm$ *0.02* | *0.000* $\pm$ *0.00* |
| | | NOCURL | 4.412 $\pm$ 0.09 | 0.054 $\pm$ 0.01 | 0.501 $\pm$ 0.03 | 0.000 $\pm$ 0.00 |
| | | NOCURL-U | 4.720 $\pm$ 0.10 | 0.038 $\pm$ 0.02 | 0.513 $\pm$ 0.03 | 0.000 $\pm$ 0.00 |
| | | NOTEARS | 4.221 $\pm$ 0.10 | **0.121** $\pm$ **0.02** | **0.613** $\pm$ **0.02** | 0.000 $\pm$ 0.00 |
| ER6 | 30 | COSMO | **6.643** $\pm$ **0.29** | 0.039 $\pm$ 0.02 | 0.420 $\pm$ 0.05 | **0.000** $\pm$ **0.00** |
| | | DAGMA | 6.837 $\pm$ 0.34 | **0.062** $\pm$ **0.02** | 0.465 $\pm$ 0.04 | *0.000* $\pm$ *0.00* |
| | | NOCURL | *6.820* $\pm$ *0.26* | 0.032 $\pm$ 0.02 | 0.368 $\pm$ 0.06 | 0.000 $\pm$ 0.00 |
| | | NOCURL-U | 6.947 $\pm$ 0.25 | 0.055 $\pm$ 0.02 | *0.492* $\pm$ *0.02* | 0.000 $\pm$ 0.00 |
| | | NOTEARS | 6.917 $\pm$ 0.37 | *0.056* $\pm$ *0.03* | **0.497** $\pm$ **0.04** | 0.000 $\pm$ 0.00 |
| ER6 | 100 | COSMO | **6.130** $\pm$ **0.06** | 0.010 $\pm$ 0.00 | *0.513* $\pm$ *0.02* | **0.000** $\pm$ **0.00** |
| | | DAGMA | 6.570 $\pm$ 0.09 | *0.046* $\pm$ *0.01* | 0.499 $\pm$ 0.01 | *0.000* $\pm$ *0.00* |
| | | NOCURL | *6.505* $\pm$ *0.11* | 0.018 $\pm$ 0.01 | 0.428 $\pm$ 0.04 | 0.000 $\pm$ 0.00 |
| | | NOCURL-U | 6.884 $\pm$ 0.09 | 0.020 $\pm$ 0.01 | 0.504 $\pm$ 0.03 | 0.000 $\pm$ 0.00 |
| | | NOTEARS | 6.533 $\pm$ 0.14 | **0.049** $\pm$ **0.01** | **0.532** $\pm$ **0.02** | 0.000 $\pm$ 0.00 |

Table 6: Experimental comparison on variance-normalized datasets.

| Graph | $d$ | Algorithm | NHD | TPR | AUC | Time (s) |
|---|---|---|---|---|---|---|
| ER4 | 30 | COSMO | **0.867** $\pm$ **1.01** | **0.953** $\pm$ **0.04** | **0.984** $\pm$ **0.02** | 88 $\pm$ 3 |
| | | SORTNREGRESS | 1.973 $\pm$ 0.76 | 0.890 $\pm$ 0.05 | 0.938 $\pm$ 0.03 | **11** $\pm$ **1** |
| ER4 | 100 | COSMO | **1.388** $\pm$ **0.69** | 0.917 $\pm$ 0.04 | 0.961 $\pm$ 0.03 | 99 $\pm$ 2 |
| | | SORTNREGRESS | 1.747 $\pm$ 0.85 | **0.937** $\pm$ **0.03** | **0.972** $\pm$ **0.01** | **98** $\pm$ **5** |
| ER4 | 500 | COSMO | 4.149 $\pm$ 1.14 | 0.819 $\pm$ 0.02 | 0.933 $\pm$ 0.01 | **437** $\pm$ **81** |
| | | SORTNREGRESS | **1.557** $\pm$ **0.39** | **0.942** $\pm$ **0.02** | **0.978** $\pm$ **0.01** | 3825 $\pm$ 171 |
| SF4 | 30 | COSMO | **0.300** $\pm$ **0.09** | **0.973** $\pm$ **0.01** | **0.997** $\pm$ **0.00** | 89 $\pm$ 5 |
| | | SORTNREGRESS | 0.517 $\pm$ 0.39 | 0.966 $\pm$ 0.03 | 0.987 $\pm$ 0.01 | **15** $\pm$ **1** |
| SF4 | 100 | COSMO | **0.482** $\pm$ **0.31** | **0.962** $\pm$ **0.02** | **0.991** $\pm$ **0.01** | **99** $\pm$ **3** |
| | | SORTNREGRESS | 0.765 $\pm$ 0.33 | 0.958 $\pm$ 0.02 | 0.989 $\pm$ 0.01 | 99 $\pm$ 6 |
| SF4 | 500 | COSMO | 1.566 $\pm$ 0.68 | 0.953 $\pm$ 0.02 | **0.989** $\pm$ **0.01** | **541** $\pm$ **15** |
| | | SORTNREGRESS | **0.788** $\pm$ **0.36** | **0.955** $\pm$ **0.03** | 0.986 $\pm$ 0.01 | 5373 $\pm$ 101 |

Table 7: Experimental comparison with SORTNREGRESS (Reisach et al., 2021).

| Graph | $d$ | Algorithm | NHD | TPR | Time (s) |
|-------|-----|-----------|-----|-----|----------|
| ER4 | 30 | COSMO | **0.867** ± **1.01** | **0.953** ± **0.04** | **88** ± **3** |
| | | DGT-SPMAP-JOINT | 5.793 ± 0.75 | 0.282 ± 0.12 | *91* ± *23* |
| | | DGT-SPMAX-JOINT | 6.353 ± 0.80 | *0.398* ± *0.09* | 140 ± 15 |
| | | DGT-SPMAP | 6.087 ± 0.97 | 0.290 ± 0.14 | 1829 ± 896 |
| | | DGT-SPMAX | *5.727* ± *0.82* | 0.198 ± 0.10 | 2006 ± 559 |
| ER4 | 100 | COSMO | **1.388** ± **0.69** | **0.917** ± **0.04** | **99** ± **2** |
| | | DGT-SPMAP-JOINT | 9.460 ± 0.81 | *0.190* ± *0.04* | *847* ± *297* |
| | | DGT-SPMAX-JOINT | 8.724 ± 1.92 | 0.187 ± 0.10 | 1261 ± 429 |
| | | DGT-SPMAP | 8.506 ± 0.91 | 0.149 ± 0.07 | 20200 ± 12514 |
| | | DGT-SPMAX | *8.152* ± *0.75* | 0.128 ± 0.05 | 24554 ± 13113 |
| SF4 | 30 | COSMO | **0.300** ± **0.09** | *0.973* ± *0.01* | *89* ± *5* |
| | | DGT-SPMAP-JOINT | 3.813 ± 0.35 | 0.382 ± 0.08 | **72** ± **17** |
| | | DGT-SPMAX-JOINT | 3.787 ± 0.80 | *0.624* ± *0.14* | 157 ± 34 |
| | | DGT-SPMAP | 3.820 ± 0.39 | 0.365 ± 0.11 | 2246 ± 701 |
| | | DGT-SPMAX | *3.740* ± *0.24* | 0.333 ± 0.12 | 2825 ± 1218 |
| SF4 | 100 | COSMO | **0.482** ± **0.31** | **0.962** ± **0.02** | **99** ± **3** |
| | | DGT-SPMAP-JOINT | 4.686 ± 0.73 | 0.317 ± 0.16 | 946 ± 249 |
| | | DGT-SPMAX-JOINT | 4.662 ± 0.68 | *0.351* ± *0.19* | *935* ± *296* |
| | | DGT-SPMAP | 4.686 ± 0.77 | 0.293 ± 0.19 | 12688 ± 11499 |
| | | DGT-SPMAX | *4.268* ± *0.43* | 0.137 ± 0.04 | 9812 ± 5257 |

Table 8: Experimental comparison with DAGUERREOTYPE (Zantedeschi et al., 2022).

As a further comparison, we present empirical results on our testbed of DAGUERREOTYPE in its four main configurations: joint optimization with SparseMAP (DGT-SPMAP-JOINT), joint optimization with Top-$k$ (DGT-SPMAX-JOINT), bi-level optimization with SparseMAP (DGT-SPMAP), and bi-level optimization with Top-$k$ (DGT-SPMAX). In all experiments runs we assume Gaussian noise terms, employ default hyperparameters from the original implementation[3], and fix the sparse operator parameter $K = 10$, as reported for the synthetic evaluation in Zantedeschi et al. (2022, Appendix D).

As highlighted by the original authors, joint optimization is significantly faster than bi-level optimization, which in turn takes significantly more time than COSMO on larger graphs (Table 8). On all analyzed configurations, the result highlights the scalability issues of DAGUERREOTYPE when compared to simpler optimization schemes. Further, the empirical results confirm the expectation in terms of computational complexity; COSMO has quadratic complexity on the number of nodes, while DGT-SPMAP has complexity $O(Kd^2)$ and DGT-SPMAX has complexity $O(K^2d^2)$. Further, the higher variance of the computational time results of DAGUERREOTYPE might be explained by the use of early stopping on model convergence. As expected, bi-level optimization generally improves performance of DAGUERREOTYPE but we still notice a non-negligible performance degradation compared to sparser ER2 and SF2 graphs reported in Zantedeschi et al. (2022).

## F  L2 REGULARIZATION

To ensure sparsity of the solution, most continuous structure learning approaches apply L1 regularization to the weighted adjacency matrix (Ng et al., b). As reported in Section 4, we adopt the same approach only on the direct matrix $\mathbf{H}$ to avoid influencing the priority vector $\mathbf{p}$. Apart from this, we found beneficial to also apply L2 regularization to the direct matrix $\mathbf{H}$. Intuitively, this avoids that the directed matrix "constrast" small values of the smooth orientation matrix given by arcs opposed to the partial order. In Figure 5, we offer a visualization of the acyclicity during training against the number of epochs. In particular, non-regularized matrices tend to be more acyclic (Figure 5a) and reach acyclicity later in the optimization process at parity of temperature (Figure 5b). In prac-

---

[3] https://github.com/vzantedeschi/DAGuerreotype

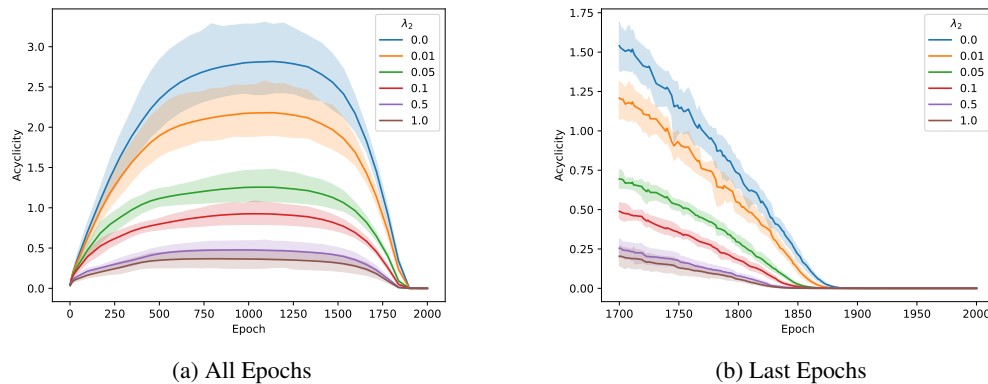

(a) All Epochs

(b) Last Epochs

Figure 5: We plot the acyclicity of the learned adjacency matrix $\mathbf{W}$ against the temperature $t$ for different regularization terms $\lambda_2$ on the directed adjacency matrix $\mathbf{H}$. Results are obtained over ten independent iterations of COSMO on an ER4 graph with Gaussian noise. In Subfigure (a) we plot the whole optimization process, while we zoom on the last epochs in Subfigure (b).

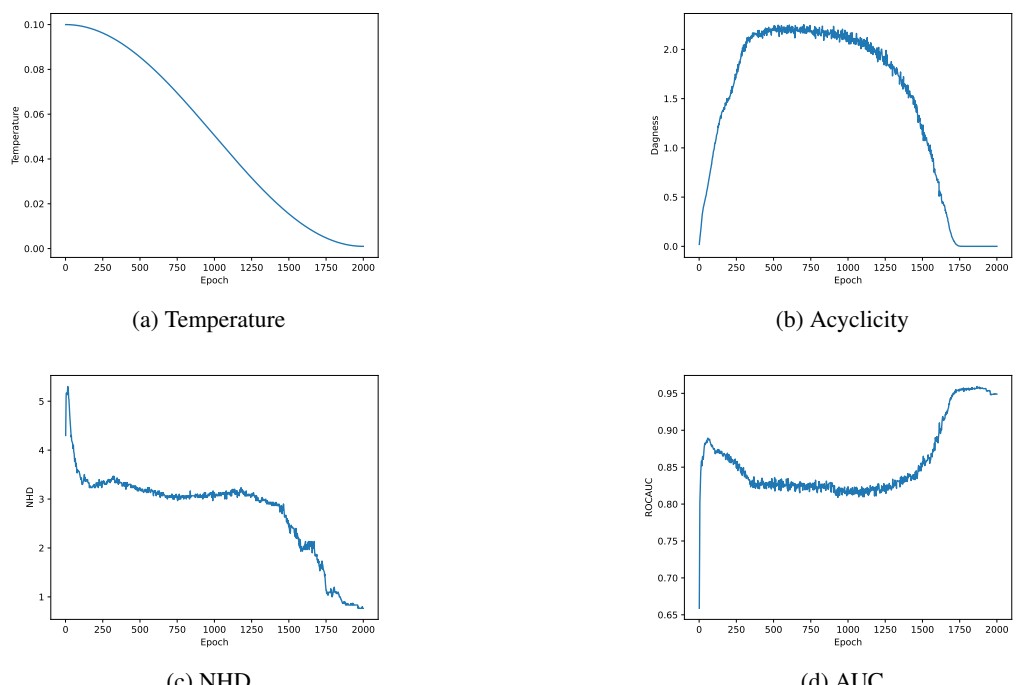

(a) Temperature

(b) Acyclicity

(c) NHD

(d) AUC

Figure 6: We report the values of (a) temperature, (b) acyclicity, (c) normalized Hamming distance, (d) and ROC-AUC against each optimization epoch of a run of COSMO on 1000 samples from an ER4 graph with 30 nodes and Gaussian noise. Notably, the model reaches an acyclic solution ($\approx$ Epoch 1700) and keeps improving before the temperature reaches zero in the last epoch.

tice, our hyperparameter search consistently awarded non-negligible L2 regularization coefficients ($\lambda_2 \approx$ 2e-2).

## G    Temperature Annealing and Model Optimization

In this section, we highlight the interaction between the temperature value and the acyclicity of the solution, together with the performance metrics on graph reconstruction. We consistently observe that the solution passes across two distinct phases (Figure 6). In all the reported metrics, after an initial convergence, the model settles with a preliminary solution. Then, after the temperature hits a sufficiently low value the acyclicity of the model rapidly decreases and the performance increase. In particular, it is worth noticing that the model reaches an acyclic solution before hitting the minimum value of the temperature and that performance metrics still vary at acyclicity zero.

In Figure 7, we propose a more detailed visualization of the evolution of the parameters during the training of COSMO, where the same phenomenom can be observed. In fact, after a first phase (Epochs 1–500) where the parameters substantially vary, the optimization is substantially stable (Epochs 500–1500) until a sufficient decrease in temperature (Epochs 1500-1700). As before, since the model converges to an acyclic solution before the temperature reaching zero (Epochs 1800–1999), we can notice that optimization continues even in this final phase, mostly increasing sparsity of the graph.

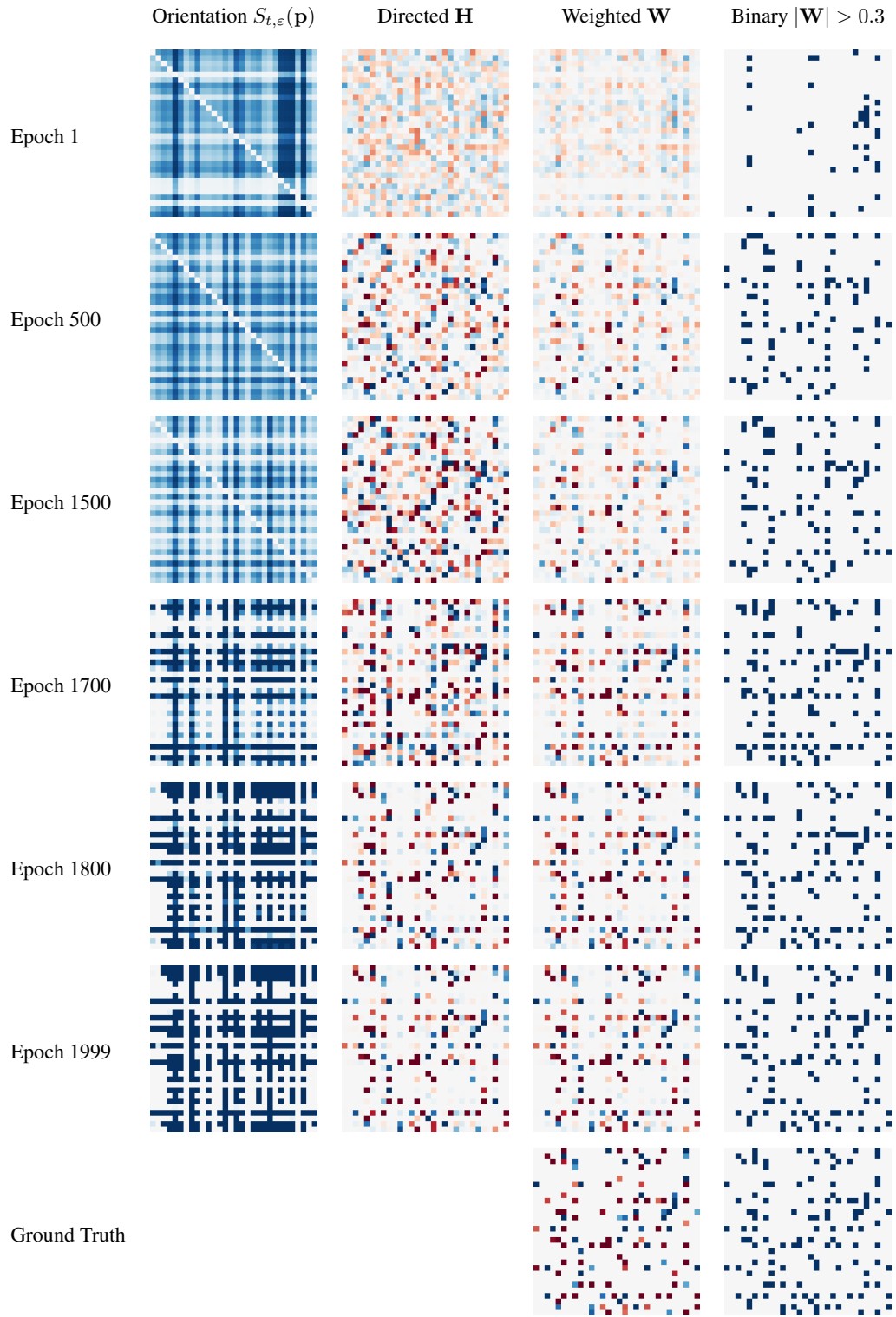

Figure 7: We report the adjacency matrices learned by COSMO at significant epochs of a run on 1000 samples from an ER4 graph with 30 nodes and Gaussian noise. As in the main body, we indicate positive values in blue and negative values in red.

