# OpenReview forum: "Constraint-Free Structure Learning with Smooth Acyclic Orientations"
_ICLR.cc/2024/Conference — ICLR 2024 poster_

### Official Review · Reviewer_t9PA · 2023-10-30

**Soundness:** 3 good
**Presentation:** 3 good
**Contribution:** 2 fair
**Rating:** 6
**Confidence:** 3

**Summary:**

The paper aims to improve the computational complexity and scalability of differentiable structure learning method. A smooth acyclicity formulation based on priority vector and tempered sigmoid are presented. With this formulation, the final approach solves an unconstrained optimization problem, requiring quadratic number of operations per optimization step. The paper presents empirical analysis to study the performance and execution time.

**Strengths:**

-   The goal of improving the computational complexity and scalability of differentiable structure learning method is well motivated.
-   The smooth acyclicity formulation and optimization scheme are sound.
-   The execution time and scalability are improved.

**Weaknesses:**

The novelty of the paper may be limited because:
- Although the paper claimed several times that it provides "the first unconstrained optimization approach that learns DAG from observational data", several existing works have done so, such as https://arxiv.org/abs/2006.10201 and DP-DAG (https://arxiv.org/abs/2203.08509). DP-DAG has been discussed in Section 2, but it may not be clear why the paper still claims being "first". Also, it would be great to compare these methods in the experiments.
- A key motivation of the work is to avoid computing the acyclicity constraint with cubic cost, which has been achieved by DP-DAG.

Furthermore, the overall idea, to my knowledge, share similarities NOCURL and is not entirely new, which has  been well acknowledged and discussed in Section 4.2 and Appendix D.2.

There is quite some performance drop for the proposed method in many cases (especially for large $d$), although the improvement in running time is notable:
- It may not be clear how to interpret AUC as compared with NHD. For instance, for the results of 500 nodes in Appendix E.3, the NHD of COSMO and DAGMA are 3.57 and 1.73 respectively, indicating that COSMO performs much worse than DAGMA (probably due to lots of false discoveries). However, the AUC of COSMO is slightly better than DAGMA.
- Therefore, I would suggest reporting NHD in the main paper that might be more commonly used in structure learning (e.g. see DAGMA and NOTEARS papers).
- It seems that there is much more performance drop for nonlinear case. Is there a reason for not including NOTEARS-MLP for 40 and 100 nodes?

**Questions:**

For some questions, see weaknesses section.

Other questions:
- Similar to NOTEARS and DAGMA, one would expect an $L_1$ regularization $\|W\|_1=\|H\circ S(p)\|_1$. Instead, Eq. (12) involves three different regularizations $\|H\|_1$, $\|H\|_2$, and $\|p\|_2$, two of which are $L_2$ regularizations.  This seems counter-intuitive and involves many hyperparameters. Can the authors explain why $\|W\|_1=\|H\circ S(p)\|_1$ would not work?
- In practice, since the temperature cannot be annealed to exactly 0, will the solution be exactly an acyclic graph before thresholding?
- Does the characterization in Eq. (7) cover the entire space of weighted DAGs $W$?
- It seems that the references provided for Section 2 "NOBEARS" and Section 1 "does not sacrifice theoretical guarantees of asymptotic convergence to acyclic solution ..." are not correct?

---

> ### Author Response · Authors · 2023-11-15
>
> We thank the reviewer for appreciating our main goal and the soundness of our proposal. We took particular care in considering their insightful comments, questions, and concerns, which we will now gladly address.
>
> **Contribution Claim.** In line with yours and other reviewers' comments, we have adjusted our claim. Further, as stated in the Related Works, we only considered for our empirical evaluation approaches that ensure the acyclicity of the solution, while GOLEM requires to iteratively threshold the solution to ensure acyclicity. Given the reviewer's suggestion, we included in Appendix E.17 an empirical comparison with GOLEM to further motivate our claim. Concerning the comparison with DP-DAG, even if possibly less apt for practical applications, simpler and scalable optimization schemes like ours are more suitable to be integrated as components in continuous causal discovery approaches. Given the reviewer's concern, we further clarified the issue in the Related Works and reported additional experiments in Appendix E.19. The additional experiments report results for DAGuerrotype (Zantedeschi et al., 2023), a similar but faster method comparable to DP-DAG, which however already highlights significant scalability issues when compared to optimization-schemes such as COSMO.
>
> **AUC, NHD, and Acyclicity** By following NOTEARS, most methods perform a thresholding step to prune off small weights and spurious arcs (Ng et al., 2023). To compare with them we adopted the same practice to report NHD, which can be computed only on binary matrices. Overall, we consider AUC to be a more suitable metric since it does not require thresholding the matrix and better captures the tradeoff between true positive and false positive ratio. Notably, the same reasoning underlies the metric chosen for validating DP-DAG in Charpentier et al. (2022). Concerning the reviewer's question, our solution is guaranteed to be acyclic before the thresholding step. To further back this claim, we updated the manuscript with a new section in Appendix G, which reports the trend of temperature and acyclicity and shows how the model reaches acyclicity zero even for non-zero temperatures.
>
> **Non-Linear.** We agree with the reviewer that there is indeed a performance drop in the non-linear scenario, which is however shared by the baselines and mainly due to the increased difficulty of the task itself. We did not include NOTEARS-MLP for larger graphs because of the significantly higher computational time required for already 20 nodes in the non-linear scenario.
>
> **Regularization.** L1 regularization favors feature selection and sparsity that, while fundamental for the overall weighted adjacency matrix $\mathbf{W}$, should not affect the priority vector. Therefore, we apply L1 regularization only on the directed matrix $\mathbf{H}$. Further, we employ L2 regularization to avoid breaking the ordering induced by the priority vector through the smooth orientation matrix. Given the concern expressed by the reviewer, we further motivated our choice by plotting in Appendix F the acyclicity during training for different values of L2 regularization. Finally, the priority vector regularization aims to avoid far apart priorities which could lead to zero-gradient issues. As we discuss at the end of Section 5.3 by reporting an ablation study, priority regularization is beneficial in particular for smaller graphs.
>
> **Space of DAGs.** As stated in Theorem 1 and proved in Appendix A.2, the formulation of Equation 7 covers all and only weighted DAGs.
>
> **NOBEARS References.** There was indeed an error in referencing Fang et al. which we fixed in the updated manuscript. Thanks for the catch!

---

> > ### Comment · Reviewer_t9PA · 2023-11-22
> >
> > Thanks for the detailed reply and additional experiments, which have addressed most of my comments. I have adjusted my score accordingly.

---

### Official Review · Reviewer_LvdG · 2023-10-30

**Soundness:** 1 poor
**Presentation:** 4 excellent
**Contribution:** 3 good
**Rating:** 5
**Confidence:** 5

**Summary:**

The paper proposes a new continuous relaxation of the acyclicity constraint for learning DAGs that is based on learning a priority score per node (that implies a partial ordering of the nodes) and relaxing its corresponding binary directed adjacency matrix to obtain non-zero gradients.
Similar to NoCurl (Yu et al. 2021), a node is an ancestor of another iff its priority score is smaller than the priority score of the other node minus a margin $\epsilon$ (that allows to represent also partial orderings). This Heaviside function is then relaxed by means of a tempered sigmoid, allowing for optimization by gradient descent.

**Strengths:**

1. The paper tackles an important problem of interest to the ICLR community by proposing a novel method for learning DAGs from observational data.

2. The overall complexity of the optimization is quadratic in the number of nodes (which is one of the lowest among DAG learning methods) allowing for scaling to graphs of $> 10^4$ nodes in the experimental analysis. The empirical analysis shows that this lower complexity translates in practice to competitive training times.

3. The paper is overall well presented. In particular, the proofs of appendix A and B are sound and easy to follow.

**Weaknesses:**

In order of gravity:

1. (major) The paper neglects the order-based literature for DAG learning which is closely related to this work. Only VP-DAG (Charpentier et al. 2021) is mentioned, but a considerable line of research exists, starting from the seminal works [1,2], to the most recent ones [e.g. 3,4] based on permutation matrices, and in particular [5] that also makes use of a priority vector. The paper does not highlight the pros and cons wrt this literature nor compare with any of the order-based methods empirically.

2. (major) Experiments are carried out on synthetic data with weight settings that are known to have trivial solutions [Reisach et al. 2021]. Hence they are insufficient to validate the method. The experimental analysis should be carried out on more challenging settings (e.g., smaller edge weights) where the trivial baseline sortnregress [Reisach et al. 2021] is not state-of-the-art.

3. By using the tempered sigmoid at training, the adjacency matrix is never acyclic. Acyclicity is only obtained in the limit of the temperature tending to 0, that however turns the gradients to 0. This implies that careful temperature scheduling is required in order to converge to a good solution.

4. The method relies on edge thresholding at test time to remove spurious relationships. However a method for choosing such threshold is not presented.

[1] Nir Friedman and Daphne Koller. Being bayesian about network structure. A bayesian approach to structure discovery in bayesian networks. Machine learning, 50, 2003

[2] Marc Teyssier and Daphne Koller. Ordering-based search: A simple and effective algorithm for learning bayesian networks. In UAI, pp. 548–549. AUAI Press, 2005.

[3] Chris Cundy, Aditya Grover, and Stefano Ermon. Bcd nets: Scalable variational approaches for bayesian causal discovery. Advances in Neural Information Processing Systems, 34, 2021.

[4] Ming Gao, Yi Ding, and Bryon Aragam. A polynomial-time algorithm for learning nonparametric causal graphs. In Advances in Neural Information Processing Systems, 2020

[5] Zantedeschi, V., Franceschi, L., Kaddour, J., Kusner, M. J., & Niculae, V. (2023). DAG Learning on the Permutahedron. In ICLR, 2023

### Update after discussion
I increased my score to 5, to acknowledge the improvements made to the paper during the discussion. Still I cannot recommend acceptance as is for the following reasons:
1. The results in the main text are still exclusively on possibly flawed settings. Because of this, these results show that the proposed method is much faster than the considered baselines, but they are not conclusive regarding its ability to perform structure learning. The authors added results on normalized setting in the appendix, however without reporting running times and only for a subset of the noise and graph configurations. The additional comparison with sortnregress, that would highlight the triviality of certain settings, is also reported separately from the main results and it again covers only a subset of the noise and graph configurations.
2. After discussion, the authors added the literature on order-based methods to their related work and reported a comparison with one of them (DAGuerreotype) in the appendix. However, this comparison is again carried out on trivial settings. Moreover the authors justified not thoroughly comparing with this line of works because they are supposedly not scalable. However one of the VP-DAG variants is one of the few methods with quadratic complexity (like the proposed method) so it is not clear how the authors can make this claim without empirical evidence.

**Questions:**

(minor) In page 15, the wrong theorem is reported.

---

> ### Author Response · Authors · 2023-11-15
>
> We thank the reviewer for acknowledging the importance of the task, appreciating the scalability of our proposal, and most importantly for their insightful comments. We hope to have addressed all the raised concerns in our following comments and in the updated manuscript in which we highlighted in red all additions and modifications.
>
> **Related Works.** Given your suggestions and the comments by other reviewers we have carefully enriched the related works section, which is now available in the updated version of our manuscript. Overall, we propose a parameterization and an optimization strategy to continuously model the space of DAGs, strictly following the line of work of NOTEARS. On the other hand, both DP-DAG (Charpentier et al. 2021) and BCD-Nets (Cundy et al. 2021) are Bayesian causal discovery algorithms that learn a distribution over DAGs given observational data. Similarly, DAGuerreotype (Zantedeschi et al. 2023) retrieves a DAG by fitting a distribution over permutations. We added and discussed the necessary references to clarify the scope of our work to the readers. For this reason, while our main empirical evaluation still focuses on other optimization schemes, we also added in Appendix E.19 empirical results of DAGuerreotype on the same experimental testbed. Our results highlight that DAGuerreotype does not scale on the size of the graph as simpler optimization schemes, which could be then more easily integrated as components in other continuous causal discovery approaches.
>
> **Evaluation.** As we remarked in the main body, we are aware that continuous approaches are particularly susceptible to data normalization and might exploit variance ordering between variables, as shown by Reisach et al. 2021 and originally by Loh and Bühlmann (2014). Therefore, empirical results on simulated datasets that do not explicitly control this condition might not generalize to real-world scenarios. However, we propose a loss-agnostic optimization scheme that aims at being integrated as a building block for causal applications as it is the case for NOTEARS, DAGMA, or NOCURL. By comparing on the exactly the same test bed of these methods, we show that at parity of conditions with the existing literature on acyclic optimization we get comparable results for significantly less time. While surely interesting as a future direction, defining function classes, appropriate losses, and identifiability results are characteristics of a causal discovery method, which is out of the scope of the current work. However, the suggestion by the reviewer is indeed extremely valuable to match our remarks with experimental results. Therefore, we include in Appendix E.18 a comparison for our and the remaining models on variance normalized data. Our results highlight how performance degradation is mostly due to the choice of MSE and does not depend in general on the particular optimization strategy, as COSMO is affected comparably to all competing methods.
>
> **Temperature.** We agree that temperature scheduling is fundamental in our solution. In practice, as we describe in the methodological section, we simply apply cosine annealing from $t=0.45$ to a relatively small temperature during training. In practice, because of the introduced regularization, the acyclicity of the solution consistently drops before reaching zero. For a visualization of this behavior please refer to the newly introduced Appendix G, where we report the trends of acyclicity and temperature. In particular, in the last phase of training, for non-zero temperature the acyclicity already reaches zero and both the directed and the smooth orientation are still optimized in this "acyclic regime".
>
> **Thresholding.** While L1 regularization already encourages sparsity, all mentioned methods prune weights with absolute values smaller than 0.3, which is a de facto standard threshold used by NOTEARS, DAGMA, NOCURL, GOLEM, and others (Ng et al. 2023). While we do not delve into this matter, we report in our experimental comparison how existing work criticizes this practice, e.g., Xu et al. (2022). Further, we emphasized in the main body the AUC metric, since it does not depend on a fixed threshold and allows us to evaluate discovery for a varying threshold. Notably, DP-DAG shares the same choice of metric.

---

> > ### Comment · Reviewer_LvdG · 2023-11-20
> > **unsolved major issues**
> >
> > I thank the authors for their reply.
> >
> > **Temperature.** Thank you for the additional results on the evolution of acyclicity during training. I find them convincing, as they show that even for non-null temperature values, the graph can be acyclic as the directed matrix H can have edge weights equal to 0. I consider this solved.
> >
> > **Thresholding.** I agree this is a weakness shared with many methods in the literature and that evaluating AUC somehow alleviates it. As such, I consider it as a minor issue.
> >
> > **Evaluation.** Evaluating the proposed method on non-trivial settings is a matter of methodological soundness and not of causal identifiability, which I agree is out of scope here. When looking at the principal results, which I take are the ones of the main text, COSMO clearly comes out as the fastest method but on tasks where a much simpler and equally fast baseline might work as well or better (sortnregress [Reisach et al. 2021]). Given that the triviality of the considered synthetic settings is recognized by the community and by the paper itself, the current comparison is inconsequential. I recommend reporting in the main text results on non-trivial settings (e.g., using smaller edge weights or standardized data) or at least compare with sortnregress to prove that the considered settings are not trivial.
> >
> > **Related Works.**
> > 1. I would like to point out that the related work has been incorrectly described in the revision of the paper:
> > _In the context of differentiable approaches, of particular interest are causal discovery methods that avoid acyclicity constraints by fitting a posterior distribution over orderings (Cundy et al., 2021) or permutations (Charpentier et al.,2022; Zantedeschi et al., 2022)._ (Cundy et al., 2021) works on permutation matrices (as Charpentier et al. 2022) while (Zantedeschi et al., 2022) works on orderings and does not fit a posterior.
> >
> > 2. Could you elaborate on why you consider these works as out of the scope of the paper? Like COSMO, they are unconstrained methods, that _continuously model the space of DAGs_, to cite your reply, learn _an acyclic graph without trading off on the adjacency matrix rank or the exactness of the acyclicity constraint_, to cite the paper, and rely on variable ordering/priority.

---

> ### Author Response · Authors · 2023-11-20
>
> We thank the reviewer for the additional and valuable feedback.
>
> **Evaluation**. COSMO does not employ early stopping or any other algorithmic conditional based on the convergence. Therefore, the running time of our parameterization is not affected by the "triviality" of the task. We produce a fair comparison by evaluating all methods in the *exact* same setting in which the corresponding papers defined them, where they all should be able to perform adequately. As suggested, we already provided the required results in Appendix E.18, which show how variance-normalization equally affects ours and the competing methods and does not add further information on their comparison. We will integrate results with "sortnregress" to better clarify the limitation, which we recall being an issue of the chosen loss function and not of ours or any particular structure learning scheme.
>
> **Related Works.**
>
> 1. We thank the reviewer for the comment on our related works summary, which we propose to rephrase as "In the context of differentiable approaches, of particular interest are causal discovery methods that avoid acyclicity constraints by fitting a distribution over permutations parameterized either explicitly (Cundy et al., 2021; Charpentier et al.,2022) or by a topological ordering (Zantedeschi et al., 2022)."
> 2. As we report in the paper, "we propose a parameterization and an optimization scheme to fastly optimize acyclic adjacencies that could be then easily integrated in causal discovery methods" and we consequently focus on alternative optimization schemes instead of many of the available and more complex DAG discovery strategies. Given the reviewer's suggestion, we included results on DAGuerreotype in Appendix E.19 to back our point and highlight how scalability is a pressing issue compared to simpler optimization strategies.

---

> > ### Author Response · Authors · 2023-11-21
> >
> > We have uploaded an updated version of our manuscript.
> >
> > **Evaluation.** We included in Appendix E.18 the results of "sortnregress" on several of our settings. Since we share the exact same experimental testbed of NOTEARS, DAGMA, and NOCURL, the "sortnregress" baseline performs better on some settings as expected, while notably being slower than our approach on larger graphs. With this additional result, we hope to have further clarified that we propose a faster but equally performing strategy compared to other optimization schemes for learning acyclic graphs, without aiming to tackle the causal discovery problem. As requested by the reviewer, the manuscript now provides results on both "*more challenging settings*" and against the sortnregress baseline, in addition to an extensive comparison with other optimization schemes in their exact empirical scenario.
> >
> > **Related Works.** Concerning the first major issue identified in the original review, we would like to remark that our related works section now references the reported works, explains why they were not included in the main empirical comparison, and finally backs this choice with dedicated experiments against several DAGuerreotype configurations on four distinct datasets.

---

### Official Review · Reviewer_tdcC · 2023-11-02

**Soundness:** 3 good
**Presentation:** 3 good
**Contribution:** 2 fair
**Rating:** 6
**Confidence:** 4

**Summary:**

The paper studies the problem of DAG structure learning from a continuous score-based perspective. In order to optimize a given score over the space of DAGs, the authors propose to decompose the parameterization by a real matrix $H$ and an orientation matrix induced by a priority vector $p$, a threshold $\epsilon$ and a temperature $t$. Then, it is shown that asymptotically as $t\to 0$ the orientation matrix corresponds to a adjacency matrix of a DAG. The authors optimize $H$ and $p$ jointly and observe speed ups over SOTA methods.

**Strengths:**

The paper is overall clear and makes a fair contribution to structure learning, which is a significant problem.
The paper is also of good quality, the authors presented a comprehensive set of experiments, showcasing the competitive performance of their method.

**Weaknesses:**

The main weakness of the paper is its novelty. The re-parametrization itself is not new, that is, the idea of decomposing a weighted adjacency matrix by a real matrix and orientation matrix. Comparing to NOCurl, the change is basically from ReLU to Sigmoid. However, I should give credits to the authors for making this arguably small change work well in practice.

**Questions:**

* Consider a $p$ vector with all entries being different. Then, for some arbitrarily small $\epsilon$, it seems to me that the orientation matrix is the adjacency matrix of a fully-connected DAG. In some sense, $\epsilon$ controls the sparsity level of the output, however, this is problematic because $\epsilon$ is not something being optimized but instead fixed a-priori. Please correct me if I am wrong.

* In the paragraph about "Direct Matrix Regularization". For the values found for $H$ and $p$ after optimizing, why not simply threshold the smooth orientation matrix by a small amount so that the "small entries" are set to zero and are not affected by the possibly large values of $H$. Given that linear models with equal variances are identifiable from the score function, L2 regularization of the weights should not be necessary and I wonder if doing this affects the performance of COSMO.

* I think more details about the optimization procedure should go to the main text. In particular, the scheduling for $t$ seems very important  and it was not clear how it was set in the main text.

* Related to my point above, I might have missed this since I only briefly looked at the appendix. But the NOTEARS $h$ is not a good proxy for acyclicity as discussed by Bello et al. (2022). In fact, given that your method does not rely on $h$, why not simply threshold the smooth orientation matrix by a very small amount and check in $O(|V|+|E|)$ that the adjacency corresponds to a DAG?

* In the experiments, the runtime for DAGMA looks different to what was reported by Bello et al. (2022). These might be because of the available implementation of DAGMA is using different hyperparameters for the optimization, but I am curious what values were used in the comparison for the main text.

* I should also note a related work on learning DAGs by Deng et al. (2023) "Optimizing NOTEARS objectives via topological swaps", which obtains better performances in structure estimation although somewhat slower.

---

> ### Author Response · Authors · 2023-11-15
>
> We thank the reviewer for appreciating our contribution and our empirical evaluation. We have carefully considered their insightful questions and the identified weaknesses, which we will individually address below.
>
> **Novelty.** Considering the weakness identified by the reviewer, we acknowledge in the main body the similarities between ours and NOCURL parameterization. However, given the theoretical guarantees ensured by our smooth orientation matrix (Theorems 1 and 2), we are able to define an optimization scheme significantly different from NOCURL. As we comment in Appendix D.2, NOCURL proposes an algorithm that extracts the ordering from a solution obtained through NOTEARS, which in turn requires evaluating acyclicity during training. On the other hand, COSMO jointly learns the ordering, is entirely differentiable, and does not evaluate acyclicity at any optimization step.
>
> **Epsilon.** The reviewer is correct that with a sufficient small $\varepsilon$, the smooth orientation matrix $S_{t,\varepsilon}$ would be a fully connected matrix. However the sparsity of the overall solution is controlled also by the directed matrix $\mathbf{H}$, since the overall DAG is given by the product $\mathbf{W}=\mathbf{H}*S_{t,\varepsilon}(\mathbf{p})$. Therefore, even if an orientation would be complete, we can still learn a sparse graph given the sparsity of $\mathbf{H}$. Notably, $\varepsilon$ cannot be zero to avoid cycles between variables with the same priority.
>
> **L2 Regularization.** We thank the reviewer for the suggestion on thresholding the smooth orientation matrix during training. We experimented with a similar approach during preliminary experiments, where we used a hard threshold to test the difference between priorities in the forward pass, while keeping the sigmoid for the backward pass. However, we found it beneficial to avoid thresholding in both passes. As a further reference, we included in Appendix F an experiment on how acyclicity evolves during training for different values of L2 regularization.
>
> **Temperature and Acyclicity.** In our methodological section we report that we perform cosine annealing of the temperature during training, but we have added a pointer to the Appendix where the choice of temperature and other hyperparameters range is discussed. Further, the matrix is guaranteed to be acyclic *without* thresholding, which we only perform to remove spurious edges. We included in Appendix G a visualization of the optimization process that shows the trend of acyclicity and performance metrics against the temperature and highlights how COSMO reaches an acyclic solution even for non-zero temperatures.
>
> **Further Additions.** We added in Appendix C.4 a table reporting the hyperparameter ranges for each baseline, including DAGMA. Also, we have updated the references according to yours and other reviewers comments, please refer to the extended discussion on related works in the updated draft.

---

> > ### Comment · Reviewer_tdcC · 2023-11-22
> >
> > I thank the authors for their response. It has helped clarify my concerns. My point about the novelty of the technical contributions remains but I appreciate the efforts put into the optimization part to make the method work. Also, I continue to feel positive towards acceptance of the paper.

---

### Official Review · Reviewer_Ar2D · 2023-11-03

**Soundness:** 3 good
**Presentation:** 2 fair
**Contribution:** 3 good
**Rating:** 6
**Confidence:** 3

**Summary:**

the paper proposes a constrain-free DAG learning formulation, based on existing notears and nocurl framework. The main innovation is to use a smoothed matrix to indicate the topological order, which enables the joint optimization of the smoothed topological matrix and a weighted adjacency matrix. Results show improved accuracy, esp when the number of dimension is high.

**Strengths:**

- the paper proposes an improved DAG learning approach without constraint, which is novel

- empirical performance on the large number of variables is solid

**Weaknesses:**

- Experiment: When the number of variable is small or medium, the accuracy seems to be worse than some existing approaches. In addition, some experiment results are missing and some elaboration is needed.

- Presentation could be improved to be more scientifically sound.


Details and questions:

- Figure 3: since you showed the results with 1e4 nodes, what is the accuracy performance of these methods with large d?
- page 13 in appendix, last line: how did the denominator become $(1 + e^{-\epsilon})^n$ in the last step?
- authors seem to over claim the contributions and often make some imprecise statements. The formulation is based on the existing nocurl formulation, with $W = H \cdot f(p)$. Nocurl also learn a topological order with p along with the typical weighted adjacency matrix H. The difference is that nocurl learns topology from a preliminary solution, while the proposed COSMO optimizes them jointly with a smoothed formulation. This in itself is a worthy contribution, no need to make exaggeration statements such as claiming "the first unconstrained optimization approach" (which would be nocurl-u by authors' definition) and nocurl is "a slightly similar model". Authors should discuss the differences.
- learning topology seems to be a hard task by itself. Gradient-based approach may not do too well on it, which explains the poorer results when the number of variable is smaller. What are authors observation on the change in topology over time (by the orientation matrix)? Does it have significant change in the beginning and then barely change toward the end of optimization steps? How does the temperature annealing change the topology over time?

**Questions:**

see above

---

> ### Author Response · Authors · 2023-11-15
>
> We thank the reviewer for recognizing the novelty of our contribution but also for their constructive criticism, which led to a significant refinement of the paper and to the addition of a novel appendix section to the manuscript. We will now punctually address each concern.
>
> **Figure 3.** The goal of the experiment reported in Figure 3 is to test the average training epoch time of each method. To this end, we limited the training of each method to a small and finite and small number of epochs and computed the average duration. The results are therefore not meaningful in terms of accuracy or performance.
>
> **Proof of Theorem 3.** We thank the reviewer for the remark. There was indeed a typo which we have corrected in the rebuttal of the manuscript.
>
> **Contribution Claim.** In Appendix D.2 we discuss the differences between our parameterization and the one used by NOCURL. The section was already referenced in the related works, but thanks to the reviewer suggestion we also added a pointer from the methodological section. There, we also changed the claim in “similar model [...] with a significantly different optimization scheme” to better highlight the differences. Also, we have adjusted our claim on the primacy of unconstrained optimization models according to the reviewer’s opinion.
>
> **Topology Change.** Overall, we observed that the topology mostly changes at the beginning of the optimization process and at the end, when the temperature approaches zero. We also consistently notice that the model reaches acyclic solutions for non-zero temperatures. We back this claim by introducing Appendix G, where we report a visualization of the smooth orientation matrix, the directed matrix and the overall graph against the trends of temperature, dagness and NHD.

---

> > ### Comment · Reviewer_Ar2D · 2023-11-20
> >
> > Thank you for your responses. You mention that "topology mostly changes at the beginning of the optimization process and at the end, when the temperature approaches zero". Is there an explanation for the changes at the end of optimization process? Does it mean that the topology changes in the beginning and then changes again (maybe once) at the end?

---

> > > ### Author Response · Authors · 2023-11-20
> > >
> > > > Is there an explanation for the changes at the end of optimization process?
> > >
> > > At the end of the optimization process, as soon as the smooth orientation matrix $S_{t,\varepsilon}(\mathbf{p})$ turns acyclic, all the arcs non-respecting the partial order induced by the priority vector $\mathbf{p}$ are effectively pruned. Then, in the final optimization epochs, when the target adjacency matrix $\mathbf{W}$ is already acyclic but the temperature $t$ still non-zero, the direct acyclic matrix $\mathbf{H}$ might still have non-zero gradients and thus adjust or prune further arcs.
> > >
> > > > Does it mean that the topology changes in the beginning and then changes again (maybe once) at the end?
> > >
> > > Yes, even if the binary adjacencies vary across the whole optimization process, we consistently observe two main structural changes during the first and the last epochs, as highlighted by the trends in Figure 6 from Appendix G.

---

### Official Review · Reviewer_Rsrs · 2023-11-05

**Soundness:** 3 good
**Presentation:** 2 fair
**Contribution:** 2 fair
**Rating:** 5
**Confidence:** 3

**Summary:**

This paper introduces a constraint-free continuous optimization scheme (COSMO) to the structure learning problem that focuses on acyclic graph reconstruction. COSMO employs a differentiable relaxation of the acyclic orientation matrix, allowing it to learn a directed acyclic graph without explicitly enforcing acyclicity constraints. The key contributions include the introduction of a smooth orientation matrix, the development of COSMO, which is significantly faster than existing constrained methods, and a thorough experimental comparison highlighting COSMO's superior performance in terms of structure recovery and computational efficiency. The paper provides valuable insights and tools for more efficient and unconstrained acyclic structure learning.

**Strengths:**

- A primary strength of the method is its provision of an unconstrained continuous optimization approach for acyclic structure learning.
- The method introduces a novel, differentiable approximation known as the "smooth orientation matrix," which depends on a temperature parameter.
-  The method outperforms certain constrained approaches in terms of speed due to the quadratic number of operations needed to reconstruct the DAG.

**Weaknesses:**

-  In Figure 1, are you learning a directed graph or a partially directed graph with undirected edges to be identified later?
Please formally define the priority vector in the notation section.
- While Figure 1 appears to be an illustration of how COSMO works, it's not very clearly explained. It would be helpful to provide a proof sketch or illustration with an example in the introduction to briefly explain how your algorithm works (given this example) and why it's an unconstrained optimization method.
- Can you provide information on the computational time and sample complexity of COSMO?
- How do you guarantee the identifiability of the recovered graph, or do the results lead to some equivalence class?
- In the related work section, consider covering some classic DAG structure learning methods.
- The method relies on the temperature parameter for the sigmoid function. While annealing the temperature is shown to be effective, selecting an appropriate temperature schedule could be a challenging task, and the method's performance may be sensitive to the choice of temperature.
- The effectiveness of this approach might depend on the initial conditions. Finding suitable initializations for the priority vector and other parameters could impact convergence and the quality of the learned DAG.
- Figure 4 does not seem very necessary, and it's not very clear. Consider moving Figure 4 to the appendix and using the space to discuss the sensitivity analysis of your method's parameter choices.

**Questions:**

See above "weeknesses" section.

---

> ### Author Response · Authors · 2023-11-15
>
> We thank the reviewer for having recognized the novelty of our method, for the insightful comments and the proposed suggestions to improve our manuscript. We hope to have addressed most of the identified weaknesses.
>
> **Figure 1.** COSMO learns a directed acyclic graph as a function of a directed graph and a priority vector. The purpose of Figure 1 is to depict the parameterization that we also describe in intuitive terms in the Introduction section before summarizing our contributions. To clarify this point, in the rebuttal update of the manuscript, we remarked in the caption that the smooth acyclic orientation is a function of the priority vector and clarified the notation.
>
> **Computational Time.** In terms of computational complexity, the construction of the directed acyclic graph given the priority vector $\mathbf{p}$ and the directed matrix $\mathbf{H}$ requires a quadratic number of steps in the number of nodes. Other than complexity results, we report in each table the computation time of each model on each dataset. In terms of sample complexity, we train all models on datasets containing 1000 realizations for each graph type, in close adherence with the NOTEARS experimental testbed.
>
> **Identifiability and Classic Methods.** As we report in the background, the identifiability of the true underlying graph requires several assumptions on the function class and the loss function. What we propose is a general approach to optimize acyclic graphs, which our empirical results show to be faster while comparable in optimization with other continuous structure learning methods. The minimization of the MSE without further assumptions does not ensure *per se* particular results on the identifiability, which require a continuous structure learner, like ours, to be embedded in a more general causal discovery framework. As suggested, we remarked on this aspect in the Related Works, to better guide the reader in the context compared to classic structure learning or causal discovery methods.
>
> **Temperature and Initialization.** We agree with the reviewer that the temperature and the initial conditions are indeed sensitive hyperparameters that might affect the method. In particular, we discuss the initialization strategy of the priority vector in Section 4. We introduced in Appendix G a visualization of the optimization process during temperature annealing, which highlights how COSMO converges to an acyclic solution before reaching zero temperature while continuing to optimize the acyclic matrix. Finally, concerning Figure 4, we gave priority to yours and other reviewers' comments and will consider moving it to the appendix according to the requested additions from the discussion.

---

### Author Response · Authors · 2023-11-15

We thank all the reviewers for their efforts in carefully reading and discussing our manuscript, and for providing their insightful feedback and comments. We have done our best to address them and we wish to engage in meaningful and productive conversations. We also thank the reviewers for highlighting the important positive aspects in our submissions, agreeing that our contribution provides a novel and computationally faster unconstrained approach to structure learning. Furthermore, we thank the reviewers for appreciating our efforts in providing a comprehensive experimental analysis.

In trying to address all of your important comments, our manuscript has undergone substantial modifications and additions, which we highlighted in red. Hence, we kindly ask the reviewers to refer to the latest version of our manuscript.

We will now address all the reviews on their respective threads.

---

### Author Response · Authors · 2023-11-21
**Discussion Reminder**

We wish to remind to reviewers that we have uploaded a revision addressing their comments, which we have also extensively answered in the corresponding threads.

In particular, we have carefully considered concerns about novelty by commenting more in depth on our proposal (Section 1) and its relation with other structure learning or causal discovery approaches (Section 2). In this respect, we have followed up on the reviewers' requests by providing additional experimental results on GOLEM (Appendix E.17), DAGuerreotype (Appendix E.19), and variance-normalized datasets (Appendix E.18). Further, we performed additional experiments to justify our regularization choices (Appendix F) and visualize the overall optimization process (Appendix G), highlighting how COSMO reaches acyclic solutions even for non-zero temperature values. We have highlighted all the new inserts in red to ease the comparison with the previous version of our manuscript.

By thanking you again for your invaluable contribution to the reviewing process, we would very much appreciate to have your feedback on our responses and engage in a fruitful discussion about our contribution before the end of the corresponding period.

---

### Meta-Review · Area_Chair_6Nqn · 2023-12-08

**Metareview:**

The authors propose a new approach to differentiable structure learning, building upon existing techniques in this area. During the discussion, a concern was raised about the treatment of recent related work, which was addressed with a detailed revision. Due to this concern, this was a borderline case, but after discussion it seems the novelty of the approach is enough to push this past acceptance.

**Justification For Why Not Higher Score:**

Borderline case

**Justification For Why Not Lower Score:**

The main objection raised by reviewers was convincingly and thoroughly addressed in the revision

---

### Decision · Program_Chairs · 2024-01-16

Accept (poster)